# ForamEcoGEnIE 2.0: incorporating symbiosis and spine traits into a trait-based global planktic foraminiferal model

**Rui Ying**[1]**, Fanny M. Monteiro**[2]**, Jamie D. Wilson**[1]**, and Daniela N. Schmidt**[1]

[1]School of Earth Sciences, University of Bristol, Bristol, BS8 1RJ, UK
[2]School of Geographical Sciences, University of Bristol, Bristol, BS8 1SS, UK

**Correspondence:** Rui Ying (rui.ying@bristol.ac.uk)

**Abstract.** Planktic foraminifera are major marine calcifiers in the modern ocean, regulating the marine inorganic carbon pump, and generating marine fossil archives of past climate change. The foraminifera contain ecogroups with and without spines and algal symbionts, creating functional trait diversity which expands their ecological niches. Here, we incorporate symbiosis and spine traits into the symbiont-barren non-spinose foraminifer functional type in EcoGEnIE to represent all the extant foraminifera species. We calibrated the modelled new traits using Latin hypercube sampling (LHS)CE1 and identified the optimal model parameters from an ensemble of 1200 runs compared with global observations from core-top sediment samples, sediment traps, and plankton nets. The model successfully describes the global distribution and seasonal abundance variation of the four major foraminiferal functional groups. The model reproduces the dominance of the symbiont-obligate group in subtropical gyres and of the symbiont-barren types in the productive subpolar oceans. Global annual mean biomass and foraminifer-derived carbon export rate are correctly predicted compared to data, with biomass ranging from 0.001 to 0.010 mmol C m$^{-3}$ and organic carbon export 0.002–0.031 mmol C m$^{-2}$ d$^{-1}$. The model captures the seasonal peak time of biomass and organic carbon export but struggles to reproduce the amplitude of both in productive areas. The sparseness and uneven distribution of observations and the model's limitation in upwelling regions likely contribute to this discrepancy. Our model overcomes the lack of major groups in the previous ForamEcoGEnIE 1.0 version and offers the potential to explore foraminiferal ecology dynamics and its impact on biogeochemistry in modern, future, and paleogeographic environments.

## 1 Introduction

Planktic foraminifera are marine-calcifying zooplankton that have populated the surface ocean since the mid-Jurassic period ($\sim$ 175 Ma). They produce calcite shells (or "tests") preserved in vast amounts of sediments. These sediments provide proxy archives (e.g. $^{13}$C, $^{18}$O, Mg/Ca) which are commonly used to reconstruct past climate conditions (Tierney et al., 2020), ocean carbonate chemistry (Hönisch et al., 2012), and to study the biotic response to environmental change (Todd et al., 2020). In the modern oceans, foraminifera contribute to 23 %–56 % of the total open-ocean CaCO$_3$ export (Schiebel, 2002) alongside the other major calcifiers, such as coccolithophores (Daniels et al., 2018) and pteropods (Buitenhuis et al., 2019). However, understanding the impacts of environmental change on foraminifera and their role in the carbon cycle is challenged by their low standing stocks in the surface ocean, a (semi)lunar reproductive cycle driving abundances and difficulties in culturing to ground truth physiology (Schiebel and Hemleben, 2017). Modelling planktic foraminifera and their ecology, therefore, has a critical role in increasing and testing our understanding of their biological and ecological influence on the marine inorganic carbon cycle and their role as a paleoproxy carrier.

Significant developments of global foraminifera models have been driven by the increasing number and extent of flux and community structure observations (Siccha and Kucera, 2017; Buitenhuis et al., 2013; Sunagawa et al., 2020). Most existing models are either empirically-based or focus on selected extant species. For example, Waterson et al. (2017) built a Maxent species distribution model based on sediment core data to study the niche variability during the Last

Glacial Maximum (LGM) compared to the Holocene. Žarić et al. (2006) constructed a statistical model that correlated hydrographical factors with sediment trap abundance of 18 dominant species. Correlative models, though, are limited for extensive future projections as they assume a constant environmental niche, neglecting adaptation and acclimation (Buckley et al., 2010). In addition, niche models do not resolve biological interactions which have an important role in shaping species distribution (Anderson, 2017).

Fraile et al. (2008, 2009), Lombard et al. (2011) and Kretschmer et al. (2016, 2018) built and extended ecophysiology-based models (PLAFOM and FORAM-CLIM) to overcome these limitations. They successfully reconstructed planktic foraminifera's geographical distribution, seasonal and vertical population dynamics, and simulated distributions in different climatologies such as the LGM (Fraile et al., 2009) and future high-emission scenarios (Roy et al., 2015). Both models are species-based and therefore cannot be applied in the deeper geological record older than the Miocene (>5 Ma) (Kucera and Schonfeld, 2007) due to a high number of extinct species and cryptic taxa with unknown novel ecologies (Renaud and Schmidt, 2003). Additionally, FORAMCLIM uses experimental growth rates to simulate foraminifera abundance and does not resolve top-down controls on foraminifera biomass. To fill the model gap and to leverage the abundant foraminiferal fossil information, a mechanistic model not limited to species is needed.

Trait-based plankton models are an alternative approach focusing on organismal traits including morphological and physiological properties instead of taxonomic identities. They provide a mechanistic way to mimic the complex ocean ecology characterising the functional groups, their traits, and associated benefits and costs (i.e. trade-offs) (Zakharova et al., 2019; Kiørboe et al., 2018). Models adopting trait-based approaches have successfully reconstructed the biomass distribution of diverse marine community including cyanobacteria (Follows et al., 2007) and diazotrophs (Monteiro et al., 2010). This modelling strategy is well suited to be applied to planktic foraminifera as functional traits such as body size (Schmidt et al., 2004a), size-normalised weight (Todd et al., 2020; Barker and Elderfield, 2002), and symbiosis (Spero and Parker, 1985) are widely measured and studied. The evolution of these functional traits has been described in detail across the Cenozoic (Ezard et al., 2011).

Critical traits of planktic foraminifera include calcification, body size, and the presence and absence of spines and symbiosis. While calcification and body size are universal traits for all foraminifera, the evolution of spines and symbiosis determine the species-level discrepancies (Aze et al., 2011). Based on the presence of symbionts and spines, foraminifera can be divided into four functional groups: (1) symbiont-barren non-spinose; (2) symbiont-barren spinose; (3) symbiont-facultative non-spinose; (4) and symbiont-obligate spinose (Table 1). Roughly 19 out of the 50 modern foraminifera species are symbiotic, bearing eukaryotic

algae such as dinoflagellates, chrysophytes, and haptophytes (Takagi et al., 2019), though this important relationship is not established for all taxa. Photosynthesising symbionts provide extra energy to foraminifera in nutrient-depleted regions (LeKieffre et al., 2018; Ortiz et al., 1995; Uhle et al., 1999). Consequently, symbiotic species dominate tropical to subtropical regions, while non-symbiont species (termed as "symbiont-barren") reach high abundance in temperate and polar oceans (Fig. 3). Some symbiont-bearing taxa cannot live without their symbionts (termed as "symbiont-obligate") (Bé et al., 1982) while others are flexible (termed as "symbiont-facultative").

The presence of calcareous spine influences the foraminifera's feeding behaviour. Non-spinose foraminifera rely on rhizopodia `CE2` to capture prey. Spinose foraminifera have spines extruding from the test that increases their effective reach range and ability to active prey; this in turn increases the ability to caption more prey types and larger cell sizes like copepods (Anderson et al., 1979). Laboratory observations show that spinose carnivory `CE3` foraminifera prefer food with a high zooplankton-to-phytoplankton protein ratio (Schiebel and Hemleben, 2017). The effective encounter rate of a spinose taxon can be 3 orders of magnitude higher than non-spinose species (Gaskell et al., 2019). Roughly half of modern species are spinose, but existing models have not taken this trait advantage into consideration.

Recently, Grigoratou et al. (2019) developed the first mechanistic and trait-based 0D model (ForamEcoGEnIE 1) for the symbiont-barren non-spinose foraminiferal group and coupled it to a carbon-centric Grid-ENabled Integrated Earth system model (cGEnIE; Grigoratou et al., 2021a), a 3D Earth System Model of Intermediate Complexity (EMIC), allowing for fast computational time and widely applied to past climates including the Paleocene–Eocene Thermal Maximum (PETM; Ridgwell and Schmidt, 2010), Last Glacial Maximum (LGM; Rae et al., 2020) and Cretaceous–Paleogene (K–Pg) boundary (Henehan et al., 2019). The computational efficiency and application to a wide range of geological periods mean ForamEcoGEnIE can be used to explore foraminiferal diversity in past climates beyond the limits of other models (Ezard et al., 2011). Here, we extend the model to ForamEcoGEnIE 2.0 by resolving three more critical functional groups of planktic foraminifera by adding the traits of symbiosis and spines (the latter tested in Grigoratou et al., 2021b). This development therefore focuses on solving foraminiferal diversity rather than marine carbonate chemistry. We tuned the model by comparing it with three global observational data compilations (sediment core-tops, plankton nets, and sediment traps) and test its ability to reproduce surface biomass, organic carbon and calcite flux, and geographic distribution in the modern climate.

## 2 cGEnIE ocean and atmosphere physics

ForamEcoGEnIE is based on cGEnIE (carbon-centric Grid-ENabled Integrated Earth system model). The fast climate and ocean physics of cGEnIE are based on a coarse-resolution 3D frictional geostrophic ocean model coupled to a 2D energy-moisture-balance atmospheric model and a dynamic–thermodynamic sea-ice model (Edwards and Marsh, 2005; Marsh et al., 2011). The ocean has a $36 \times 36$ equal-area horizontal grid (uniform in longitude and sine-uniform in latitude) with 16 logarithmically spaced vertical levels as defined in Cao et al. (2009). The physical model is coupled with a model of ocean biogeochemical cycles (Ridgwell et al., 2007; van de Velde et al., 2021), sea-floor sedimentary processes (Ridgwell and Hargreaves, 2007), and marine ecosystem processes (Ward et al., 2018). The plankton ecosystem is resolved in the surface layer (0–80.8 m). The model presented in this study is configured with a seasonally forced pre-industrial climate state and an atmospheric $CO_2$ concentration restored to 278 ppm.

## 3 Size-based plankton ecosystem framework EcoGEnIE

### 3.1 Biogeochemical tracers

The model has three main state variables: inorganic resources ($i_r$), living biomass ($i_b$), and detritus ($i_d$). Each state variable contains multiple biogeochemical tracers: carbon, phosphorus, and iron. Plankton populations are counted in notation $j$, and each plankton includes the three tracers above, although autotroph plankton (phytoplankton and symbiotic foraminifera) have an extra tracer of chlorophyll (noted in Chl). Figure 1 shows a schematic of the plankton types including foraminifera and denotes elements in different colours.

### 3.2 Plankton cell size and quota

In EcoGEnIE, individual body size determines key physiological processes including nutrient uptake, photosynthesis, grazing gain, and predation through allometric scaling (West et al., 1997) because of its role as a master trait among pelagic organisms (Andersen et al., 2016). The modelled size-dependent parameters (except for photosynthesis) follow a generic power law: $P = aV^b$, where $P$ is the size-based parameter, $V$ the spherical biovolume, and $a$ and $b$ the allometric intercept and exponent, respectively.

A fundamental size-based concept of EcoGEnIE is the plankton cell quota for various elements. The carbon quota content ($Q_C$) follows the same power law as per Eq. (1). Variable stoichiometry ($Q_{i_b}$, Eq. 2) is achieved by the ratio of assimilated nutrients biomass ($B_{i_b}$, where $i_b$ stands for P, Fe, or chlorophyll) to carbon biomass ($B_C$) (Droop, 1968; Flynn, 2008). This stoichiometry limits nutrient uptake rate

($Q_{i_b}^{stat}$, Eq. 3) as per Geider et al. (1998), with a higher value close to its maximum ($Q_{i_b}^{max}$) lowering the nutrient uptake or chlorophyll synthesis rate. The nutrient quota range ($Q_{i_b}^{min}$, $Q_{i_b}^{max}$) is proportional to the carbon quota ($Q_C$):

$$Q_C = aV^b, \tag{1}$$

$$Q_{i_b} = \frac{B_{i_b}}{B_C}, \quad i_b = \text{P, Fe, Chl}, \tag{2}$$

$$Q_{i_b}^{stat} = \left( \frac{Q_{i_b}^{max} - Q_{i_b}}{Q_{i_b}^{max} - Q_{i_b}^{min}} \right)^{0.1}. \tag{3}$$

### 3.3 Plankton biomass dynamics

The biomass of any plankton group ($j$) and element ($i_b$), $B_{j,i_b}$, varies due to a combination of potential physiological processes that are determined by the type of organism: nutrient uptake, grazing gains, grazing losses, mortality, and respiration loss (Eq. 5). Foraminifera-related specific processes will be introduced in following sections. We refer readers to Ward et al. (2018) for the detailed description of EcoGEnIE that expands on the description below.

$$\frac{\partial B_{j,i_b}}{\partial t} = \underbrace{\mu_{j,i_b} \cdot B_{j,C}}_{\text{nutrient uptake}} + \underbrace{B_{j,C} \cdot \lambda_{i_b} \sum_{j_{prey}=1}^{J} G_{j,j_{prey},i_b}}_{\text{grazing gains}}$$

$$- \underbrace{B_{j_{pred},C} \cdot \sum_{j_{prey}=1}^{J} G_{j_{pred},j,i_b}}_{\text{grazing losses}} - \underbrace{m_j \cdot B_{j,i_b}}_{\text{mortality loss}} - \underbrace{r_{j,C} \cdot B_{j,i_b}}_{\text{respiration loss}} \text{TS3}.$$

$$\tag{4}$$

### 3.4 Inorganic nutrient dynamics

The inorganic resource state variable ($R_{i_r}$) varies with nutrient uptake ($V_{j,i_r}$) and dissolved inorganic carbon (DIC) with the living organisms' respiration ($r_{j,C}$):

$$\frac{\partial R_{i_r}}{\partial t} = \begin{cases} \sum_{j=1}^{J} -\mu_{j,i_r} \cdot B_{j,C}, & i_r = \text{Fe, P} \\ \sum_{j=1}^{J} -\mu_{j,i_r} \cdot B_{j,C} + \sum_{j=1}^{J} r_{j,C} \text{TS4}, & i_r = \text{C}. \end{cases} \tag{5}$$

Additional sources and sinks of nutrients such as remineralisation of organic matter and air–sea gas exchange are computed in the biogeochemical module BIOGEM (Ridgwell et al., 2007).

### 3.5 Particulate organic matter dynamics

Particulate organic matter (POM) flux ($F$) is a combination of predators' messy feeding (the first term) and the mortality

loss (the second term) from all plankton groups (Eq. 6):

$$F = \sum_{j_{\text{pred}}=1}^{J} \sum_{j_{\text{prey}}=1}^{J} (1 - \beta_{j_{\text{pred}}, i_{\text{d}}})(1 - \lambda_{j_{\text{pred}}, i_{\text{b}}}) G_{j_{\text{pred}}, j_{\text{prey}}} B_{j_{\text{pred, C}}}$$

$$+ \sum_{j=1}^{J} (1 - \beta_j) m_j B_{j, i_{\text{d}}}, \tag{6}$$

where $\beta$ `CE4` is the fraction of dissolved organic matter (DOM) subject to diffusion and advection by ocean circu-
5 lation. The remaining fraction $(1 - \beta$ `CE5`$)$ is the particulate organic matter (POM) subject to redistribution through the water column by sinking. The parameter $\beta$ is a sigmoid function depending on maximum and minimum DOM fraction $(\beta_{\text{max}}, \beta_{\text{min}})$ of predators' equivalent sphere diameter (ESD)
and the size $\beta_{\text{s}}$ at which DOM / POM ratio equals 1 (Ward and Follows, 2016). Smaller cell sizes are associated with greater proportion as DOM.

$$\beta = \beta_{\text{max}} - \frac{\beta_{\text{max}} - \beta_{\text{min}}}{1 + \beta_{\text{s}}/\text{ESD}}. \tag{7}$$

Messy feeding behaviour is modelled as the unassimilated
fraction $(1 - \lambda_{j_{\text{pred}}})$ of prey which is limited by the size-independent maximum efficiency coefficient $(\lambda_{\text{m}})$ and the nutrient limitation (Fe or P):

$$\lambda = \lambda_{\text{m}} \cdot \min\left[Q_{\text{P}}^{\text{stat}}, Q_{\text{Fe}}^{\text{stat}}\right]. \tag{8}$$

## 4 ForamEcoGEnIE 1 brief description

ForamEcoGEnIE 1 accounted for the feeding behaviour and calcification of foraminifera (Grigoratou et al., 2019, 2021a). It implemented a predator–prey interaction $(G_{j_{\text{pred}}, j_{\text{prey}}}$ `TS5`, Eq. 9) using a Holling type II model (Holling, 1965), where the overall grazing rate depends on the total available prey
$(F_{j_{\text{pred}}})$, the maximum grazing rate of predators $(G_{\text{pred}}^{\text{m}})$, and the half-saturation concentration of available food $(k_{j_{\text{prey}}})$; it is regulated by temperature limitation $(\gamma_{\text{T}})$, a prey-switching term $(\Phi)$, and a prey-refuge protection $(1 - e^{\Lambda F_{j_{\text{pred}}}})$. The other elements' biomass $(B_{i_{\text{b}}})$ are then scaled using plank-
ton's own biomass ratio $(B_{i_{\text{b}}}/B_{i_{\text{C}}}$ `TS6`$)$. The calcification trait was included by reducing foraminifera palatability $(P_{\text{p}}$ which influences $F_{j_{\text{pred}}}$ (Eq. 10) and mortality rate $(m_j$, Eq. 5) to account for higher protection against predators and infections at the expense of a lower $G_{\text{foram}}^{\text{m}}$ (Eq. 9). We also introduce the
ForamEcoGEnIE 2 parameters (spine effect $\tau$ and a mixotrophy limitation $\lambda_{\text{h}}$) here which is set to 1, i.e. not functioning in ForamEcoGEnIE 1:

$$G_{j_{\text{pred}}, j_{\text{prey}}} \text{ } \underbrace{\gamma_{\text{T}} \cdot \lambda_{\text{h}}}_{\text{limitations}} \cdot \underbrace{\frac{G_{\text{pred}}^{\text{m}} \text{ } F_{j_{\text{pred}}} \text{ }}{\tau k_{j_{\text{prey}}} \text{ } + F_{j_{\text{pred}}}}}_{\text{overall grazing rate}} \cdot \underbrace{\Phi_{j_{\text{pred}}, j_{\text{prey}}}}_{\text{Switching}}$$

$$\cdot \underbrace{(1 - e^{\Lambda F_{j_{\text{pred}}}})}_{\text{prey refuge}}, \tag{9}$$

$$F_{j_{\text{pred}}} \text{ TS20} = P_{\text{p}} \text{ TS21} \cdot B_{j_{\text{prey}}} \text{ TS22}$$

$$\cdot \exp\left[-\left(\ln\left(\frac{\mu_{j_{\text{pred}}, j_{\text{prey}}}}{\mu_{\text{opt}}}\right)\right)^2 / (2\sigma_{j_{\text{pred}}}^2)\right]. \tag{10}$$

Predators select their prey (Eq. 10) based on the predator–
40 prey size ratio $\mu_{j_{\text{pred}}, j_{\text{prey}}}$ relative to the optimal value $\mu_{\text{opt}}$, the predators' food range $\sigma_{j_{\text{pred}}}^2$, and the calcification protection $P_{\text{p}}$. Foraminifera in both ForamEcoGEnIE 1 and 2 are set as herbivores.

The grazing process like other metabolic processes in
EcoGEnIE is temperature-dependent, following the universal metabolic theory (Brown et al., 2004). The body temperature of ectothermic plankton is determined by the ambient seawater environment $(T)$. Temperature regulation $\gamma_{\text{T}}$ acts on metabolic processes including respiration, nutrient uptake,
and predation. It is modelled through an Arrhenius-like function (Eq. 12), where the parameter $A$ determines temperature sensitivity and reference temperature $(T_{\text{ref}})$ is the temperature allowing $\gamma_{\text{T}} = 1$:

$$\gamma_{\text{T}} = e^{A(T - T_{\text{ref}})}. \tag{11}$$

The prey-switching term $(\Phi_{j_{\text{pred}}, j_{\text{prey}}})$ simulates the feeding habitat of zooplankton (Eq. 9). The exponential $s$ defines the active level of zooplankton predators which capture abundant prey with higher priority when $s$ increases. Foraminifera in both ForamEcoGEnIE 1 and 2 are assumed to be ambush
passive predators with $s = 1$:

$$\Phi_{j_{\text{pred}}, j_{\text{prey}}} = \frac{\left(F_{j_{\text{pred}}} \text{ TS23}\right)^s}{\sum_{j_{\text{prey}}=1}^{J} \left(F_{j_{\text{pred}}} \text{ TS24}\right)^s}. \tag{12}$$

A refuge term $(1 - e^{\Lambda F_{j_{\text{pred}}}})$ in Eq. (9) is added to decrease the grazing rate when prey availability lowers. The coefficient $\Lambda$ determines the strength of such protection.

## 5 ForamEcoGEnIE 2: improved calcification and more functional groups

In ForamEcoGEnIE 2, we add symbiosis and spine traits for foraminifera to result in four functional groups (Table 1, Fig. 1). We also implement a new calcification energetic cost
by using a respiration term rather than a reduced maximum growth rate in ForamEcoGEnIE 1.

### 5.1 Calcification trait trade-offs

#### 5.1.1 Benefit: mortality protection

The mortality loss term for zooplankton scales with a basal
rate constant $m_{\text{b}}$ (Eq. 5). As per Grigoratou et al. (2019, 2021a), this is downscaled for foraminifera by a protection term $P_{\text{m}}$, where a lower value of $m_j$ indicates a higher protection from the foraminiferal test against viral and bacterial

**Table 1.** The four modelled functional groups of planktic foraminifera and their species representative in ForamEcoGEnIE 2.0.

| Spine trait | Symbiosis trait | Species example | Species number* | Model implementation |
| --- | --- | --- | --- | --- |
| Spinose | Symbiont-obligate | *Globigerinoides ruber* | 17 | This study |
| Spinose | Symbiont-barren | *Globigerina bulloides* | 2 TS25 | This study |
| Non-spinose | Symbiont-facultative | *Neogloboquadrina dutertrei* | 5 | This study |
| Non-spinose | Symbiont-barren | *Neogloboquadrina pachyderma* | 23 | Extended from ForamEcoGEnIE 1 |

* Count from Schiebel and Hemleben (2017).

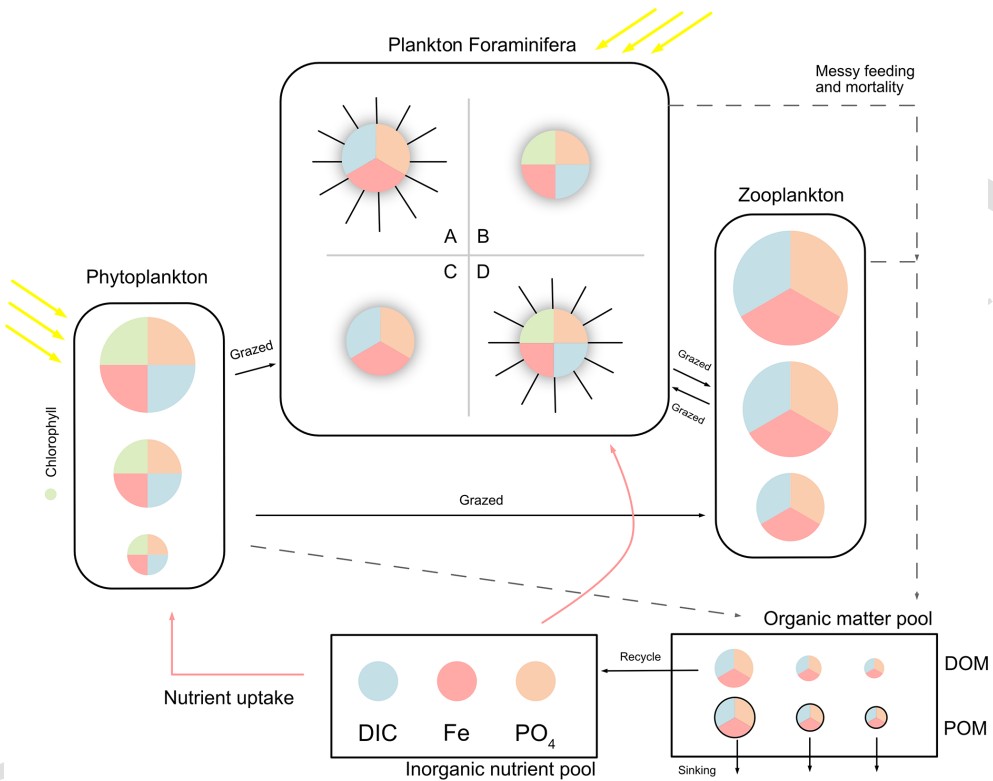

**Figure 1.** Schematic representation of the ForamEcoGEnIE 2.0 model structure. The model includes the biogeochemical cycles of C, Fe, and P (shown in different colours), various plankton size classes and four main groups of planktic foraminifera: A – symbiont-barren spinose group; B – Symbiont-facultative non-spinose group; C – symbiont-barren non-spinose group; D – symbiont-obligate spinose group. DIC stands for dissolved inorganic carbon and PO$_4$ for phosphate. The model represents nutrient uptake (red arrows), dissolved and particulate organic matter production (DOM and POM) caused by messy feeding and mortality (dashed arrows), and zooplankton grazing (black arrows).

infections:

$$m_j = P_m^{\text{TS26}} \cdot m_b. \tag{13}$$

#### 5.1.2 Benefit: protection from predators (palatability)

As per ForamEcoGEnIE 1.0, calcification protects from grazing and is defined by $P_p$ which reduces the biomass loss from predation (Eq. 10).

#### 5.1.3 Cost: higher metabolic cost

We modified the metabolic cost of calcification defined in Grigoratou et al. (2019, 2021a) by replacing the original reduced maximum growth rate (or specifically maximum grazing rate) with a temperature-dependent respiration loss term. We made this change because (1) extra respiration is a more biologically realistic cost with (2) this temperature-dependent term reconciling the model with the low-latitude biomass observation. The respiration $r_j$ present in Eq. (5) scales with carbon biomass and is multiplied by constant $r_b$ and temperature limitation (Eq. 11). We assumed that the lost carbon from respiration is instantly recycled back to dissolved inorganic carbon (DIC) pool:

$$r_j = r_b \cdot \gamma_T. \tag{14}$$

## 5.2 Spine trait trade-offs

Spines are an important part of foraminiferal taxonomy. Spines, like the overall test, are made of calcite. A range of biological functions are assumed to be linked to symbiosis and feeding behaviour (Schiebel and Hemleben, 2017).

### 5.2.1 Benefit: enhanced grazing

Studies show that spinose foraminifera are more efficient in capturing and digesting prey due to the spine and rhizopodia networks (Anderson and Bé, 1976). Spines widen the prey availability of immotile foraminifera and facilitate the capture of larger preys. Non-spinose species cannot hold active prey and only accept smaller particles of copepods in the laboratory observations (Anderson et al., 1979; Hemleben et al., 1989). Grigoratou et al. (2021b) modelled such benefit by reducing the half-saturation constant (conventionally noted as $k$ TS27 in a Michaelis–Menten model). Here, we adopt this approach and reduce $k_{j_{\mathrm{prey}}}$ by multiplying a scaling parameter $\tau$ ($0 < \tau < 1$; Eq. 10).

### 5.2.2 Other trade-offs as calcification: higher metabolic cost and reduced palatability

We assume that the metabolic cost and protection from the spines are characterised the same way as for calcification (Eqs. 13–14). Spinose foraminifera have a higher cost for calcification due to the slightly higher amounts of carbonate and a stronger protection than non-spinose taxa (Table 2). We did not reduce the mortality term as this was not supported by direct evidence.

## 5.3 Symbiosis trait trade-offs

Symbiosis is a novel trait in the model, commonly seen in marine organisms including foraminifera. Many planktic foraminifera harbour algae (e.g. dinoflagellate, diatom) within their cells (Takagi et al., 2019). We represent these symbiotic species in the model as a single organism which combines heterotrophy and autotrophy, equivalent to a calcifying mixotroph. We use the trait-based representation of mixotrophy of Ward and Follows (2016), where any plankton can "naturally" predate and photosynthesise. While mixotrophs have this ability in the model, this is turned off for the rest of plankton (i.e. $V_{\mathrm{m}}$ is 0 for zooplankton and $G_{\mathrm{m}}$ is 0 for phytoplankton).

### 5.3.1 Benefit: enabled autotrophy for planktic foraminifera

The symbiont has a cell size that is defined via a symbiont / foraminifera size ratio $\psi$ (Eq. 16) to characterise the symbiont's affinity in taking up nutrients and light. Photomicrograph observations showed that foraminifera symbionts are about 1 : 20 smaller in size than the host cell (Takagi et al., 2019):

$$V_{\mathrm{s}} = \psi^3 V_{\mathrm{h}}. \tag{15}$$

The generic nutrient uptake of symbionts follows a Michaelis–Menten function limited by mixotrophy ($\lambda_{\mathrm{s}}$), quota ($Q_{i_{\mathrm{r}}}^{\mathrm{stat}}$), and temperature ($\gamma_{\mathrm{T}}$), where the variable ($R$) represents nutrient resources. The half-saturation constant is replaced by nutrient affinity, a more mechanistic parameter for nutrient uptake $\alpha_{i_{\mathrm{r}}}$ TS28. Nutrient affinity is often referred to "clearance rate" and regarded as a proxy of competitive strength (Fiksen et al., 2013). According to Edwards et al. (2012) review on phytoplankton trait trade-offs, nutrient affinity is negatively related to cell size because of lower surface to volume ratio, while the maximum uptake rate ($V_{\mathrm{m}}$) is positively related:

$$\mu_{i_{\mathrm{r}}}\,{}^{\text{TS29}} = \lambda_{\mathrm{s}}\,{}^{\text{TS30}} \cdot Q_{i_{\mathrm{r}}}^{\mathrm{stat}}\,{}^{\text{TS31}} \cdot \gamma_{\mathrm{T}} \cdot \frac{V_{i_{\mathrm{r}}}^{\mathrm{m}}\,{}^{\text{TS32}}\alpha_{i_{\mathrm{r}}}\,{}^{\text{TS33}} R\,{}^{\text{TS34}}}{V_{i_{\mathrm{r}}}^{\mathrm{m}}\,{}^{\text{TS35}} + \alpha_{i_{\mathrm{r}}}\,{}^{\text{TS36}} R\,{}^{\text{TS37}}}. \tag{16}$$

The symbionts' photosynthesis growth is modelled following a size-dependent unimodal equation (Geider et al., 1998; Moore et al., 2001). This equation has higher explanatory power for eukaryotic phytoplankton cells than a power law (Bec et al., 2008). The maximum photosynthesis rate $P_{\mathrm{C}}^{\mathrm{m}}$ is determined by dimensionless parameter $P_a$, $P_b$, $P_c$ and the biovolume of symbiont $V_{\mathrm{s}}$, and the mixotrophy cost $\lambda_{\mathrm{s}}$:

$$P_{\mathrm{C}}^{\mathrm{m}}\,{}^{\text{TS38}} = \frac{\lambda_{\mathrm{s}}\,{}^{\text{TS39}}\left(P_a + \log_{10} V_{\mathrm{s}}\,{}^{\text{TS40}}\right)}{P_b + P_c \log_{10} V_{\mathrm{s}}\,{}^{\text{TS41}} + \log_{10} V_{\mathrm{s}}^2\,{}^{\text{TS42}}}. \tag{17}$$

The practical photosynthesis rate is further constrained by nutrient availability (the smallest between $\gamma_{\mathrm{Fe}}$ and $\gamma_{\mathrm{P}}$), temperature ($\gamma_{\mathrm{T}}$), and light intensity ($\gamma_{\mathrm{I}}$):

$$P_{\mathrm{C}} = P_{\mathrm{C}}^{\mathrm{m}} \cdot \min\left[\gamma_{\mathrm{P}}, \gamma_{\mathrm{Fe}}\right] \cdot \gamma_{\mathrm{T}} \cdot \gamma_{\mathrm{I}}.\,{}^{\text{TS43}} \tag{18}$$

Nutrient limitation $\gamma_{i_{\mathrm{r}}}$ ($i_{\mathrm{r}}$ is either P or Fe, see the definition in Eq. 2) is determined by the minimal value of the phosphorus or iron limitation term, which follows the quota relationship in Droop (1968):

$$\gamma_{i_{\mathrm{r}}}\,{}^{\text{TS44}} = \frac{1 - Q_{i_{\mathrm{r}}}^{\mathrm{min}}/Q_{i_{\mathrm{r}}}\,{}^{\text{TS45}}}{1 - Q_{i_{\mathrm{r}}}^{\mathrm{min}}/Q_{i_{\mathrm{r}}}^{\mathrm{max}}\,{}^{\text{TS46}}}, \qquad i_{\mathrm{r}} = \mathrm{Fe, P}. \tag{19}$$

Light limitation follows the model of Moore et al. (2001), where $I$ represents light intensity, $\alpha$ is initial slope of the photosynthesis rate–light intensity curve limited by Fe content ($\gamma_{\mathrm{Fe}}$), and $Q_{\mathrm{Chl}}$ is chlorophyll quota.

$$\gamma_{\mathrm{I}} = 1 - \exp\left(\frac{-\alpha \cdot \gamma_{\mathrm{Fe}} \cdot Q_{\mathrm{Chl}} \cdot I}{P_{\mathrm{C}}^{\mathrm{m}} \cdot \gamma_{\mathrm{T}} \cdot \min\left[\gamma_{\mathrm{P}}, \gamma_{\mathrm{Fe}}\right]}\right)\,{}^{\text{TS47}} \tag{20}$$

### 5.3.2 Cost: downgrading autotroph and heterotroph efficiency

The cost of mixotrophy is that both autotrophic and heterotrophic processes (i.e. photosynthesis and grazing rates) are scaled down (by multiplying factor $\lambda_s$ and $\lambda_h$ for symbionts and hosts, respectively: $0 < \lambda_s, \lambda_h < 1$, Eqs. 9 and 16) compared to the pure autotroph or heterotroph specialist (Castellani et al., 2013; Våge et al., 2013; Ward and Follows, 2016). We distinguish between symbiont-obligate and symbiont-facultative foraminifera using different $\lambda_s/\lambda_h$ parameter values to reflect their different dependency on symbionts (Table 2).

### 5.4 Approximating foraminiferal calcite export using fixed PIC/POC ratio

Planktic foraminifera produce organic carbon in the subsurface water column (Salter et al., 2014) and sequester inorganic carbon into the deep oceans via their dead tests (Schiebel, 2002). The organic carbon flux derived from foraminifera is treated the same way as in EcoGEnIE as discussed in Sect. 3.4. The calcite export, specific to foraminifera, is approximated by multiplying the foraminiferal bulk organic carbon export with a globally uniform particle inorganic carbon (PIC) to organic carbon (POC) molar ratio of 0.36 based on the empirical data by Schiebel and Movellan (2012).

## 6 Model parameterisation

### 6.1 Plankton community size structure

We resolve eight size classes of phytoplankton, seven size classes of zooplankton, and one size class for each of the four foraminiferal groups. Phytoplankton and zooplankton size classes include 0.6, 1.9, 6.0, 19.0, 60.0, 190.0, 600.0, and 1900.0 μm, with zooplankton missing the smallest class due to minimum prey size. While the size structure of these plankton is fixed, we tested the foraminiferal ESD ranging from pre-adult (60 μm) to adult (600 μm) using the ensemble described below. Each test contains one randomly assigned foraminiferal size and this is same for each foraminiferal group. However, we found that the size (190 μm) from a previous study (Grigoratou et al., 2021a) still achieved the best score (Table 2).

### 6.2 Experiments with sampled parameters

We run an ensemble of 1200 model experiments, each testing a different combination of parameter values (Table 2), to explore all possible trait values and select the best trait combinations to match available foraminiferal observations (Sect. 3.2). The parameter sets are generated using a Latin hypercube sampling (LHS) algorithm that samples values of 12 foraminiferal parameters from uniform parameter distributions (Table 2; Sarrazin et al., 2016). However, several rules are set in the sampling: (1) the spinose ones always own higher palatability and mortality protection (and corresponding respiration cost) than the non-spinose ones; (2) the symbiont size for both symbiont-facultative and symbiont-obligate groups are set to the same value; (3) all the foraminifera have the same size. Each simulation is run for 250 years continuing from a 10 000-year spin-up simulation as the ecosystem structure typically reaches equilibrium after ∼ 50 years. The other ecosystem parameters are the same as Ward et al. (2018) (Table S3).

### 6.3 Observations for comparison

We used multisource data compilations (sampled from sediment core-tops, sediment traps, and plankton nets) to calibrate the model (see below). To simplify the model–data comparison, we assume that the biomass/export changes between the pre-industrial age (the model) and the present climate represented in plankton net and sediment trap samples are negligible. This is because (1) planktic foraminifera have stably low biomass; and (2) sediment trap and plankton net data were collected over a wide time range (1970s–2010s) with changing climatologies.

### 6.3.1 Relative abundance

We used a sediment core-top census data compilation (Siccha and Kucera, 2017) to represent a long-term mixed Late Holocene baseline (pre-industrial) to validate the spatial abundance patterns of each modelled foraminiferal group. We calculated the modelled relative abundance of each group based on its carbon export production.

To determine the observed relative abundance, we compiled species into functional groups using species traits defined by Schiebel and Hemleben (2017) and Takagi et al. (2019) (Table S4). We regridded the observations into the model grid (averaging each data grid point onto the cGEnIE grid). We ignored species with less than 3 % local abundance (a few specimen) to avoid uncertainties caused by transport via ocean currents (van Sebille et al., 2015) and taxonomic uncertainties of rare taxa. This threshold is determined by the standard error of Fisher's diversity index (Fisher et al., 1943). We used the Mielke measure (details in Sect. 6.4) to quantify the model–data fit. We omitted the Arctic Ocean and the Mediterranean Sea in the model–data comparison because the model resolution in these regions is too low to represent adequate ocean physics.

### 6.3.2 Annual average biomass and export

To validate the modelled biomass and organic carbon export, we compiled two global datasets: (1) plankton net data from the first 100 m (if sampled, otherwise the nearest depth that is no more than 120 m) for biomass and (2) sediment trap data

**Table 2.** List of the foraminifera-relevant model parameters tested in the global sensitivity analysis (GSA) and identified optimal parameter values for each foraminiferal group.

| Related trait(s) | Parameter | Description | Tested range[a] | Unit | Optimal values (bn[b]) | Optimal values (bs[b]) | Optimal values (sn[b]) | Optimal values (ss[b]) |
|---|---|---|---|---|---|---|---|---|
| Foraminiferal size | ESD | Equivalent sphere diameter | [60, 600] | μm | **190[c]** | 190 | 190 | 190 |
| Calcification/spine | $p_m$ | Protection from mortality | [0–1] | | **0.6** | 0.6 | 0.6 | 0.6 |
| | $p_p$ | Protection from grazing | [0–1] | | **0.8** | 0.7 | 0.8 | 0.7 |
| | $r$ | Respiration rate | [0–0.02] | mmol C d$^{-1}$ | **0.04** | 0.06 | 0.04 | 0.06 |
| Spine | $\tau$ | Coefficient of grazing half saturation | [0–1] | | / | **0.9** | / | 0.9 |
| Symbiosis | $\psi$ | Symbiont to foraminiferal size ratio | [0–0.05] | | / | / | **0.0015** | 0.0015 |
| | $\lambda_s$ | Symbiont autotroph efficiency | [0–1] | | / | / | **0.2** | **0.8** |
| | $\lambda_h$ | Foraminiferal heterotroph efficiency | [0–1] | | / | / | **0.8** | **0.55** |

[a] All scaling parameters are sampled from values of 0 to 1; respiration terms are as per Ward et al. (2018); the symbiont cell size ratio upper bound is calculated from Takagi et al. (2019). For any other plankton group where these traits are not relevant, scaling parameters are set to 1 and cost parameters are set to 0. [b] bn – symbiont-barren non-spinose; bs – symbiont-barren spinose; sn – symbiont-facultative non-spinose; ss – symbiont-obligate spinose. [c] The bold parameters are also shown in other groups with same trait(s).

for carbon export. We converted the units of plankton net ("number m$^{-3}$") and sediment trap data ("number m$^{-2}$ d$^{-1}$") into "mmol C m$^{-3}$" and "mmol C m$^{-2}$ d$^{-1}$" using the empirical factor of 0.845 μg C per specimen from Schiebel and Movellan (2012). We converted modelled carbon export production (mmol C m$^{-3}$ d$^{-1}$) into "mmol C m$^{-2}$ d$^{-1}$" multiplying it by the surface-layer depth (80.8 m) to compare with sediment-trap-generated export observations. The full list of plankton net and sediment trap data sources is in Tables S1 and S2.

Both datasets are classified by species and were regridded into the model resolution following the methods of the core-top data. We calculated the annual average at each grid point to remove seasonality and interannual variability. However, the plankton nets are mostly sampled within 1 month (Fig. S1) and represent a day's snapshot, such that the annual mean biomass is likely overestimated as the nets would be typically sampled during higher production times. In contrast, sediment traps are deployed over 6 months or more (Fig. S1), thereby capturing seasonal variation. Sediment traps were deployed at different depths, typically over 1000 m and thereby deeper than our surface layer. We assume that sediment trap depth has negligible impact on foraminiferal export because foraminiferal tests sink relatively fast due to large size (Takahashi and Be, 1984; Caromel et al., 2014).

### 6.3.3   Seasonality

To complement the annual comparison, we analysed the modelled seasonal pattern by finding each group's first month with peaking production. We also provided a comparison with plankton net and sediment trap data for most sampled locations in the Supplementary Material. We did not attempt

to calculate the Mielke measure (Sect. 6.4) CE6 for seasonal model–data comparisons because (1) the temporal coverage of observations is too low at most locations, and (2) the number of available locations is insufficient, creating large spatial bias towards specific oversampled locations.

### 6.4   Model performance metrics

We used the Mielke measure, or M-score (Watterson, 1996; Watterson et al., 2014), to quantify the model–data fit in (1) relative abundance and (2) annual average biomass/carbon export (Eq. 21). This metric is a non-dimensional transformed mean square error combining mean and variance information (Gregoire et al., 2011; Hemer and Trenham, 2016). The score spans from $-1$ (low performance) to 1 (high performance) with 0 representing no predictive skill, and negative values representing negative correlation:

$$M = \frac{2}{\pi} \arcsin \left[ \frac{\sum_{i=1}^{n} (M_i - O_i)^2 / n}{\sigma_m^2 + \sigma_o^2 + (\mu_m - \mu_o)^2} \right]. \qquad (21)$$

The numerator corresponds to the mean square error, with $M_i$ and $O_i$ denoting the model and observational value in the $i$th grid point, respectively, and $n$ the total number of grid points. The variance and mean are respectively denoted as $\sigma^2$ and $\mu$, with superscripts m and o representing model and observed fields, respectively.

### 6.5   Global sensitivity analysis

We conduct a global sensitivity analysis (GSA) to explore the model robustness of our 1200 experiments using the PAWN method (Pianosi and Wagener, 2015). This method measures the sensitivity of model outputs (focusing on the M-score here) to different values of 12 input parameters (shown in

Table 2). A total M-score is calculated by summing scores of each foraminiferal group in biomass, POC export, and relative abundance (i.e. the total score ranges from $-12$ to 12). To further measure the uncertainty and robustness of the GSA results, we also apply a bootstrapping method with 1000 resamples. This approach enables us to understand the confidence intervals of the sensitivity indices without running more experiments (Wagener and Pianosi, 2019). We bootstrapped our data using the rsample CE7 package (Frick et al., 2022 TS48) in the R software environment v4.1 (R Core Team, 2021).

## 7 Model results

### 7.1 Model ensemble results

The 1200-member ensemble shows the ability to reproduce the observed POC export and relative abundance in terms of spatial pattern and values (both with the highest total M-score $>1.0$) but struggles with capturing the observed biomass (total M-score $<0.5$) (Fig. 2). The M-score heatmap (Fig. 2) shows that the model runs cluster into four groups when compared to the three observational datasets. Cluster C, covering most parameter combinations, has an overall low performance in predicting foraminiferal metrics. Cluster D shows an inverse abundance distribution compared to the observation. Cluster B only predicts POC export. Cluster A achieves the highest (i.e. the best) relative-abundance M-score with good predictions for biomass and POC export. Cluster A is also the only cluster with low foraminiferal export, suggesting that low export is associated with parameter values required to have a high total M-score. The sensitivity analysis confirms this, as model performance is sensitive to those parameters controlling the source/sink of foraminiferal export: symbiont size ($\psi$), autotrophy efficiency ($\lambda_s$), and palatability reduction ($P_p$) (Fig. 3). Models with low export production and higher M-scores tend to have smaller foraminiferal size and symbiont-to-host size ratio (for symbiotic groups) that facilitates the survival of foraminifera in the low-nutrient regions like subtropical gyres. These runs in cluster A also tend to have less than 20 % decreased palatability caused by the shell and a high respiration cost, driving low biomass and export (Fig. S2). In contrast, the runs with negative scoring (Cluster D) have larger foraminiferal size and higher protection against grazing (Fig. S3). These results suggest that foraminiferal body size and the calcification trait have a crucial role in foraminiferal distributions to achieve a match to the observed data. Questions addressing the size trait in more detail, like life history and geographic size distribution (Schmidt et al., 2004b), cannot be answered with this model ensemble as all foraminiferal groups are assigned the same narrow cell size per run, even though they vary between runs.

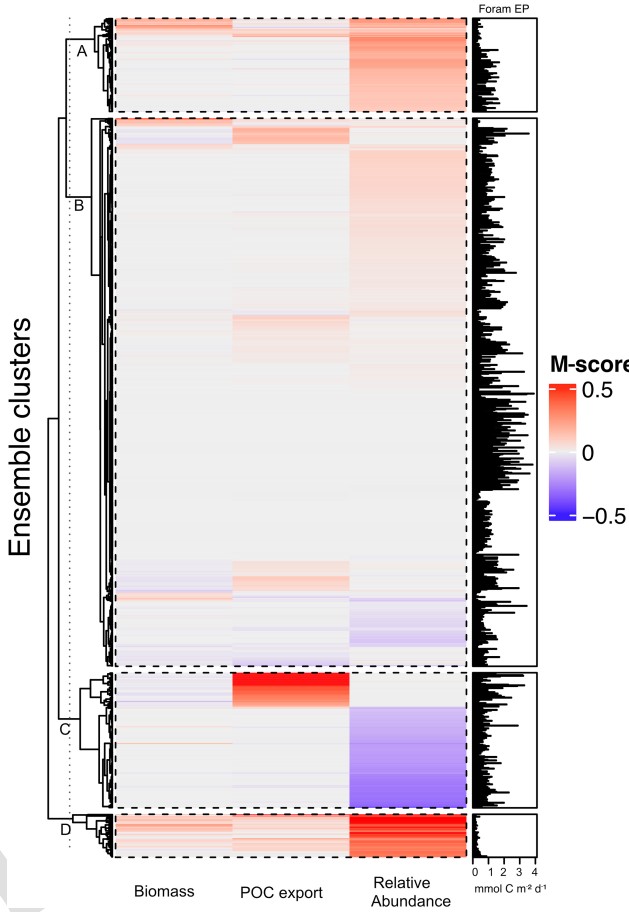

**Figure 2.** Foraminiferal M-score heatmap of the model ensemble for foraminiferal biomass (plankton net data), POC export (sediment trap data), relative abundance (sediment core-top data). Each of the first three columns shows the M-score sum of the four foraminiferal groups, and the fourth column shows the sum of the left three columns. The right panel shows the global annual mean export production of all foraminiferal groups. The ensemble cluster was derived from a complete linkage clustering algorithm (Legendre and Legendre, 1998 TS49). Higher M-scores have a better performance against observations, whilst negative values stand for negative correlation.

We selected the model with the parameter set (Table 2) that leads to the highest total M-score (Table 3), hereafter termed the optimal model. This optimal model also has the highest M-score for the relative abundance (group mean $= 0.3$) for each group (Fig. S5) and POC export (group mean $= 0.16$; Fig. S4). More details are discussed in the next sections.

### 7.2 Relative abundance distribution of foraminiferal groups

Our optimal model run compares generally well with the core-top data showing the relative spatial distribution of the four foraminiferal functional groups (Fig. 4; Table 3). The

**Table 3.** M-score values across foraminiferal groups for the optimal parameter set. The total foraminiferal M-score is the sum of the M-scores of the four functional groups.

| Groups | M-scores | | | | |
| --- | --- | --- | --- | --- | --- |
| | Symbiont-barren non-spinose | Symbiont-barren spinose | Symbiont-facultative non-spinose | Symbiont-obligate spinose | Total foraminifera |
| Biomass | 0.19 | 0.08 | −0.05 | −0.05 | 0.17 |
| POC export | 0.11 | 0.07 | 0.43 | 0.02 | 0.63 |
| Relative abundance | 0.51 | 0.35 | 0.02 | 0.32 | 1.20 |
| Row sum | 0.81 | 0.50 | 0.40 | 0.29 | 2.00 |

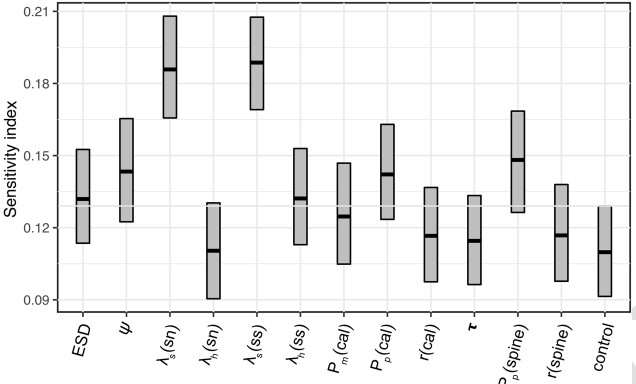

**Figure 3.** Model parameter sensitivity for overall model performance (summed M-scores). Bar boundaries indicate the 95 % confidence interval with the thick line showing the mean value. The grey line indicates the non-influential upper limit of the index value as control group. sn – symbiont-facultative non-spinose; ss – symbiont-bearing spinose. cal – the abbreviation of calcification; $\tau$ is the spine effect on grazing rate.

model agrees with the presence/absence of most groups in the sediment trap and plankton net studies (Figs. 5 and 6). The model suggests that the symbiont-obligate spinose group is the most abundant with a global abundance of 60.7 % (Fig. 4g), dominating the tropical open oceans. In contrast, the symbiont-barren non-spinose (Fig. 4a) and spinose groups (Fig. 4c) dominate in the mid-to-high latitudes, contributing 25.5 % and 9.4 % of the global foraminiferal abundance, respectively (note that the symbiont-barren spinose type contains a small number of taxa with a relatively high contribution to the abundance). The model underestimates the symbiont-facultative group (Fig. 4e) with visible model–data disparities in the eastern equatorial pacific where the sediment data show high abundance. This discrepancy may be due to the resistance to dissolution of some species (e.g. *N. dutertrei*) in high productivity settings as suggested in a previous model study (Lombard et al., 2011). Importantly though, it is not very clear what triggers the presence or absence of symbionts, why this relationship changes and often the taxa are less well studied. The fact that the summed abun-

dances of these two symbiotic groups agree with the observations indicates the ability of the symbiont-facultative group to exploit the same benefits as the symbiont-obligate one. It also highlights our need to better understand how often symbiosis is used by the former group and what triggers the switch to the loss of symbionts.

Overall, the modelled root mean square error (RMSE) of relative abundance varies between 12 % and 42 % (Table S5). This result is comparable to the previous species-based models, like FORAMCLIM (5 %–23 %, Lombard et al., 2011) and PLAFOM (22 %–25 %, Fraile et al., 2009), which rely on well-established foraminiferal species observations. Our results affirm that symbionts and spines and their assumed trade-offs are sufficient to explain significant parts of the relative abundance's geographic distribution. The distribution of non-symbiotic foraminifera in the model follows the biogeography of the prey abundance with high numbers in high-nutrient areas (Fig. S6). In contrast, symbiotic foraminifera grow in low-nutrient areas because they have small-sized symbionts with high nutrient affinities. The model underestimates symbiont-barren spinose foraminifera (mainly *G. bulloides*) in the Arabian Sea and South China Sea (Fig. 4c, d TS50), probably because the model does not include their carnivorous feeding strategy.

### 7.3 Annual average biomass of foraminiferal groups

The model reproduces low biomass in planktic foraminifera in agreement with the plankton net data (Fig. 5). The global annual mean biomass ranges from 0.001 to 0.010 mmol C m$^{-3}$, equivalent to 0.08–0.8 mmol C m$^{-2}$, with the largest contribution from the symbiont-barren non-spinose group (Fig. 7). Integrating across all groups, the model estimates a global foraminiferal biomass of 6.83 Tg C (Fig. 7). Our annual mean biomass estimate is within the MAREDAT project result (0.24–0.94 mmol C m$^{-2}$) (Schiebel and Movellan, 2012).

The optimal model M-score is low for biomass when compared to the plankton net tow ($<0.2$; Table 3), possibly because of the low data coverage and the previously mentioned intrinsic seasonal bias in the data compared to annual averages. Our ensemble resulted in higher M-scores for biomass,

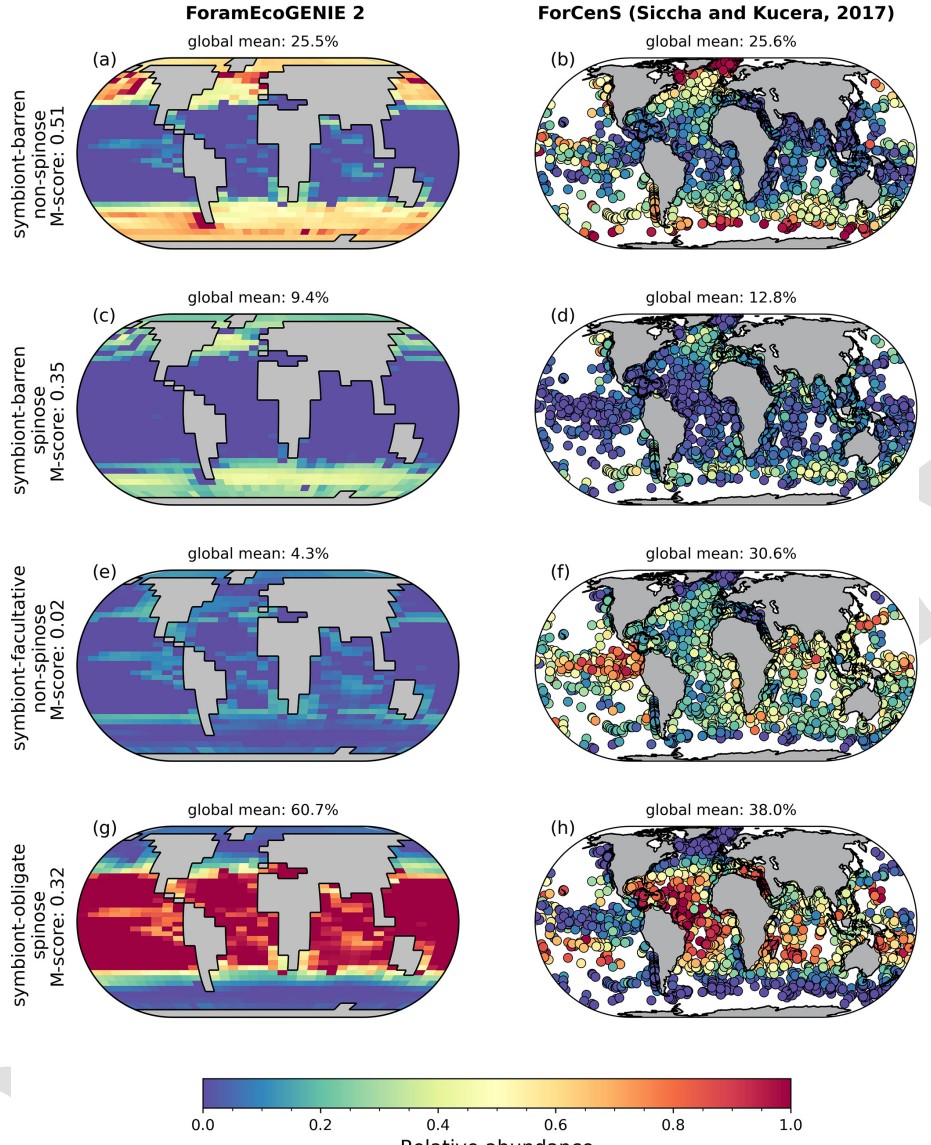

**Figure 4.** Relative abundance of the four modelled (**a, c, e, g**) planktic foraminiferal function groups, compared to the ForCenS sediment core-top dataset (**b**, **d**, **f**, **h**; Siccha and Kucera, 2017). Subplot titles show the M-scores derived relative to observations and the global mean of relative abundance.

but at the cost of a lower M-score for relative abundance and export.

## 7.4 Annual average POC and calcite export of foraminiferal groups

The model reproduces consistent distributions and magnitude of POC export compared to sediment trap data for all four groups (Figs. 6 and 7). The model estimates a POC export of $0.002$–$0.031\,\mathrm{mmol\,C\,m^{-2}\,d^{-1}}$, which agrees well with $0.001$–$0.026\,\mathrm{mmol\,C\,m^{-2}\,d^{-1}}$ for the sediment trap data, despite a medium total M-score for the model POC ex-

port (0.63) likely caused by the limited geographic coverage akin to the biomass comparison.

Globally, the model suggests a foraminifera-derived organic carbon export of $0.1\,\mathrm{Gt\,C\,yr^{-1}}$, dominated by the symbiont-barren non-spinose group (55 %), followed by the symbiont-barren spinose, symbiont-facultative, and symbiont-obligate groups (30 %, 3 %, and 11 %, respectively). Integrating across the ecogroups and using the empirically averaged PIC:POC ratio of 0.36 (Schiebel and Movellan, 2012), our model estimates a total calcite flux of pelagic foraminifera of $0.033\,\mathrm{Gt\,PIC\,yr^{-1}}$ (Fig. 8). This model estimate is at least 5 times smaller than Schiebel (2002)'s estimate of $0.16$–$0.39\,\mathrm{Gt\,PIC\,yr^{-1}}$ (for the top 100 m). There are

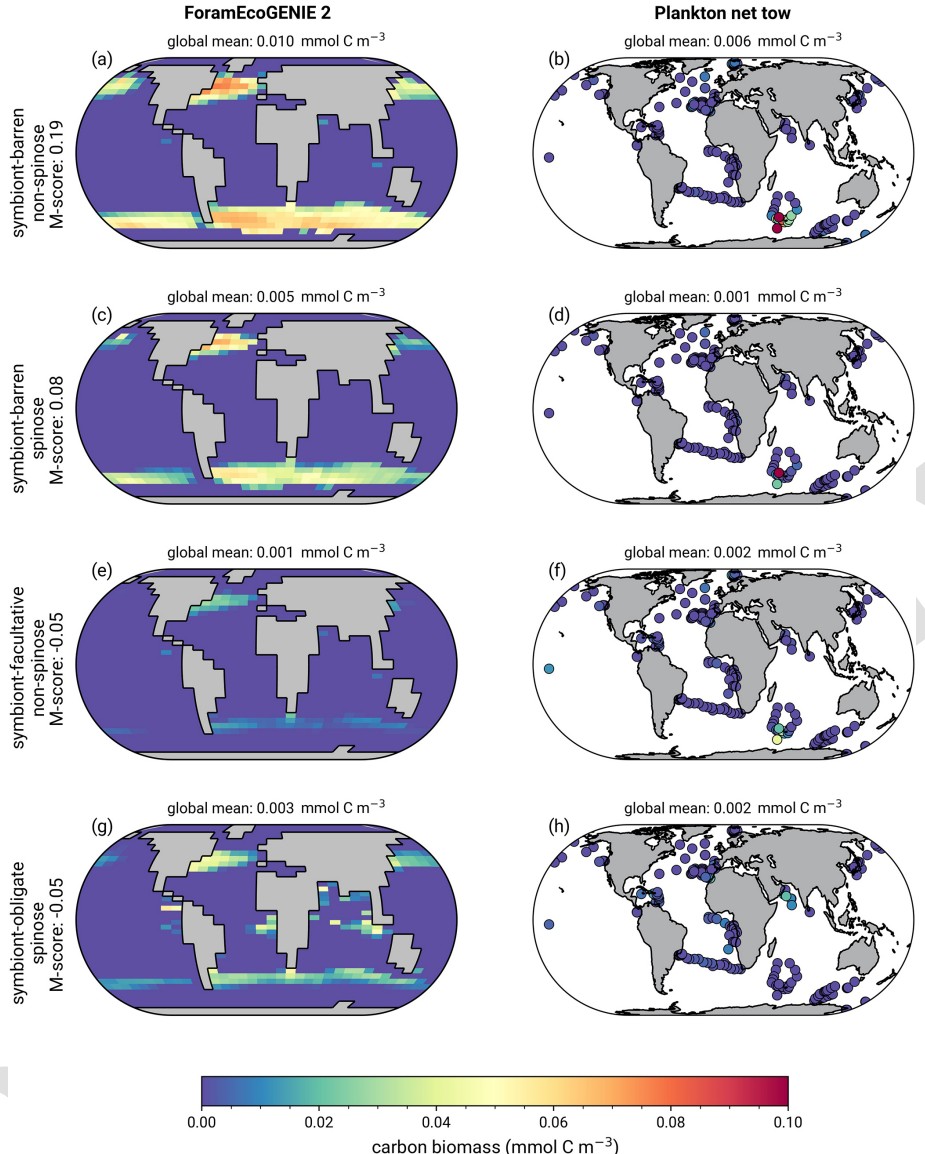

**Figure 5. (a, c, e, g)** ForamEcoGEnIE 2.0 annual average foraminiferal biomass (mmol C m$^{-3}$) compared with plankton net data (**b, d, f, h**) for the four main functional groups of planktic foraminifera.

a number of possible reasons for this: (1) a field site selection bias to avoid regions which have very low abundance, (2) our calibration of modelled surface export with deep sediment traps data characterised by typically lower export (as deployed at about 2 km depth), and (3) the temporal variability of observation which is not well captured in the model.

## 7.5 Seasonal variations of foraminiferal biomass and POC export

Our model shows the different seasonal patterns for each foraminiferal group (Fig. 9), generally consistent with sediment trap study (Jonkers and Kučera, 2015). Jonkers and Kučera (2015) divide the foraminiferal assemblages into a warm group (representing the symbiont-bearing group), cool and temperate group (representing the two symbiont-barren groups), and deep-dwelling group according to their seasonal cycle patterns. The cool/temperate group blooms in spring or summer (Fig. 9a), while the warm group in tropical oceans shows weak and random seasonality (Fig. 9d). The model also captures the earlier-when-warmer signature in the cool/temperate group, i.e. the peaking time is strongly coupled to temperature gradient from high to low latitude (Fig. 9a).

The model generally underestimates seasonal amplitudes of export production (Fig. S7). Plankton net data cannot be compared seasonally due to the very short nature of data collection, despite the general agreement (Fig. S8). The low

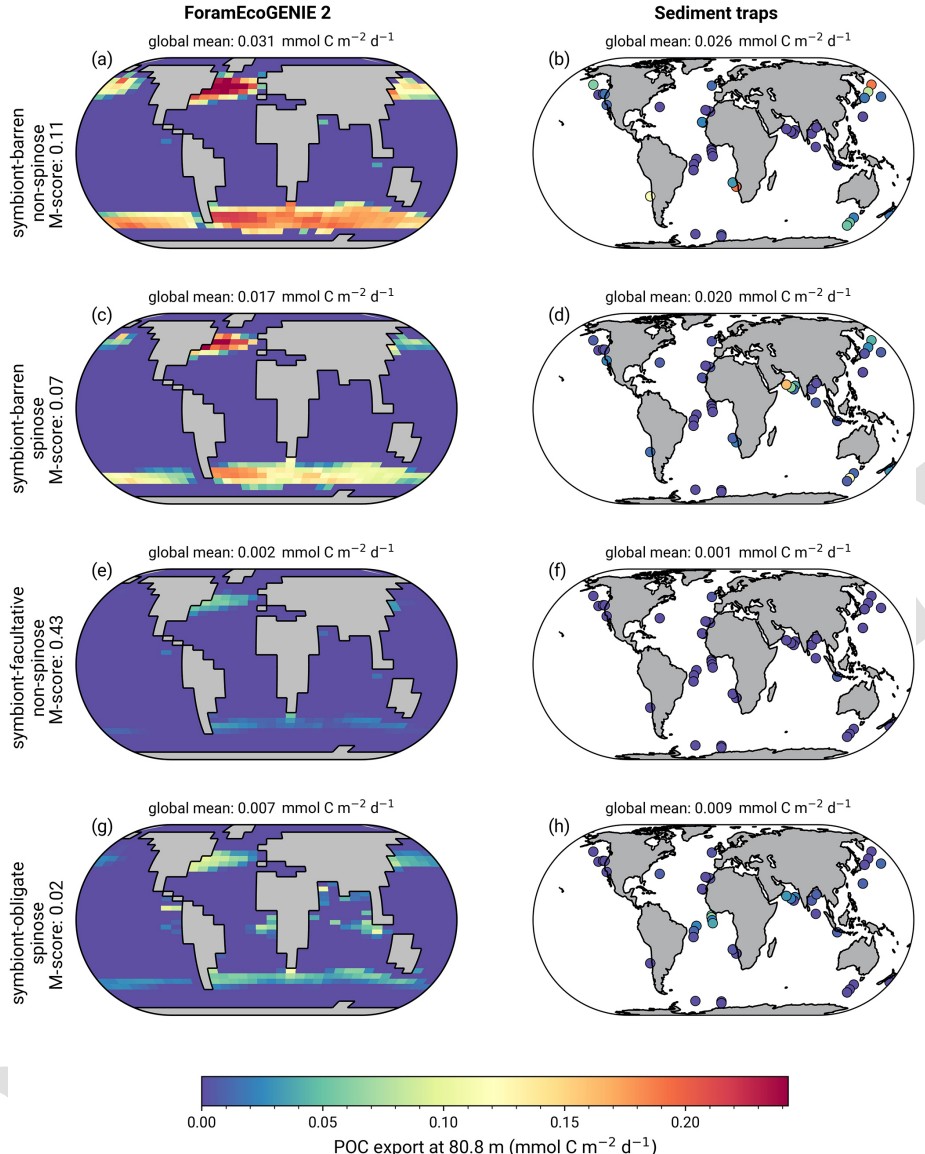

**Figure 6. (a, c, e, g)** ForamEcoGEnIE 2.0 foraminiferal annual average POC export (mmol C m$^{-2}$ d$^{-1}$) below the euphotic zone (80.8 m) in comparison to sediment trap samples **(b, d, f, h)**. TS51

model export production is not unique to our model and also evident in PLAFOM 2.0 (Kretschmer et al., 2018). Intra-annual variabilities in abundance are driven by the seasonal environmental changes which determine how optimal foraminifera are in the ecological niche. While temperature is often assumed as the primary driver for foraminiferal ecology (Schmidt et al., 2004b; Be and Hamlin, 1967), many other parameters such as primary productivity are correlated with temperature and hence difficult to separate their effects (Jonkers and Kučera, 2015). We suggest that additional functional trait data collections assessing temporal variability, increased geographic coverage, information on deeper-dwelling species, and information on life history traits will contribute to resolve this gap in the future. CE8

## 8 Comparison to prior model iterations

By comparing ForamEcoGEnIE 2.0 with EcoGEnIE (Ward et al., 2018) and ForamEcoGEnIE 1.0 (Grigoratou et al., 2021a), we find that adding foraminiferal groups increases the total plankton mean body size in the tropical and subtropical regions by roughly 20 % due to the larger size of foraminifera (modelled as 190 μm, Fig. 10c). At the same time, the new iteration decreases the plankton mean size in subpolar areas (<10 %) due to additional grazing pressure by foraminifera on the plankton. In contrast, the total plankton biomass stays almost the same between the model versions because of the low standing stocks of foraminifera. Net primary productivity (NPP) and POC export also do not change,

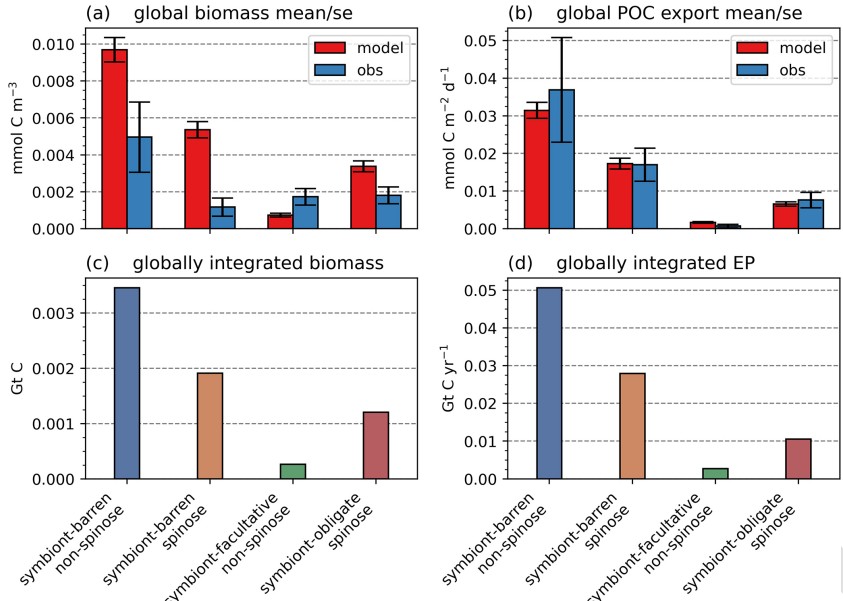

**Figure 7.** Global ForamEcoGEnIE 2.0 annual average biomass and POC export produced by the four foraminiferal groups: **(a)** modelled (red) and observational (blue) biomass (mmol C m$^{-3}$); **(b)** POC export below the euphotic zone (mmol C m$^{-3}$ d$^{-1}$). Bar height and error bar represents the spatial mean value and standard error, respectively. Panels **(c)** and **(d)** show the globally integrated model estimates for **(c)** carbon biomass (Gt C) and **(d)** export production (EP, Gt C yr$^{-1}$).

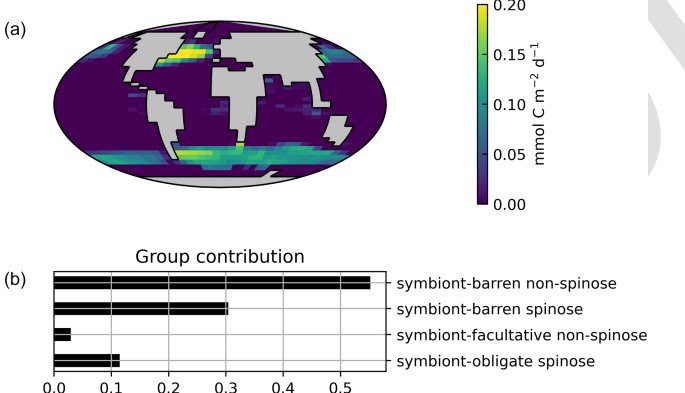

**Figure 8.** Global ForamEcoGEnIE 2.0 estimates for **(a)** surface foraminiferal calcite flux (at 80.8 m; mmol C m$^{-2}$ yr$^{-1}$) and **(b)** groups contribution.

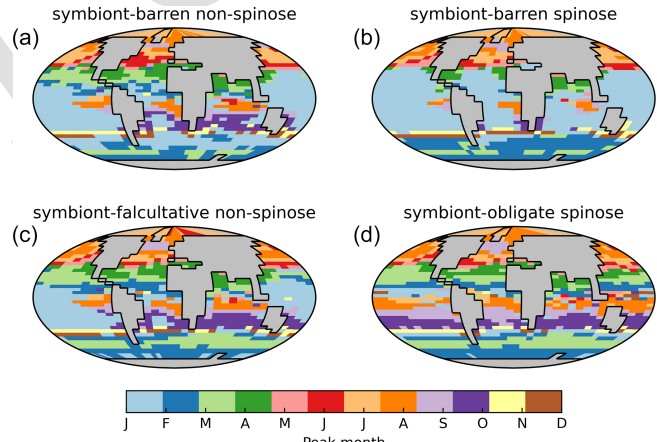

**Figure 9.** The peak month of modelled biomass annual time series of each foraminiferal group in our best ForamEcoGEnIE 2.0 run. Note that the months in Southern Hemisphere indicate the opposite seasonality of the Northern Hemisphere.

apart from a small drop in the subpolar regions due to enhanced foraminiferal grazing. Therefore, ForamEcoGEnIE 2.0 performs as well as the previous version in terms of total plankton size, biomass, carbon export, and NPP, while capturing foraminiferal diversity and biogeography.

While ForamEcoGEnIE 2.0 developments focused on improving diversity in plankton ecology, it also lays the foundation for future studies on the ocean carbon cycle under different climates, past or future. For example, the inclusion of spinose foraminifera is important for particle sinking as they produce and export more calcite than their non-spinose

counterpart (Takahashi and Be, 1984). It also opens the door for studies of past climates by expanding the foraminiferal global niche, which may influence the ocean carbon cycles by changing the locations of calcite export and distribution of surface alkalinity. So far, no Earth system model has included foraminiferal groups acting on the carbon cycle, which would be an important avenue for future research.

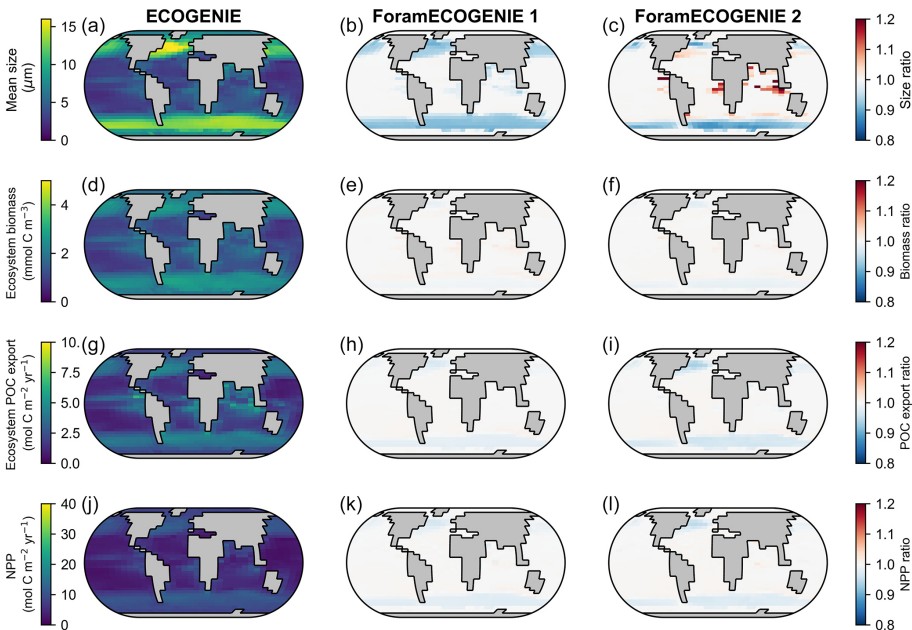

**Figure 10.** Comparison of the tuned ForamEcoGEnIE 2.0 (third column, with four foraminiferal groups) with EcoGEnIE (first column; Ward et al., 2018) and ForamEcoGEnIE 1.0 (second column, with non-spinose non-symbiont foraminifera only; Grigoratou et al., 2021a) for **(a–c)** total plankton mean size, **(d–f)** total plankton biomass, **(g–i)** total POC export, and **(j–l)** net primary production (NPP). The first column displays absolute values while the latter two show the ratio relative to the first column.

## 9 Model limitation

While making explicit progress in including planktic foraminifera into a modelling framework with a range of traits, ForamEcoGEnIE 2.0 is limited by the non-explicit implementations of spines and symbiosis. Currently, our model represents symbiosis based on mixotrophy implementation. According to the definition of mixotrophy types in Mitra et al. (2016), our modelling approach falls within constitutive (inherent or innate) mixotrophy rather than the non-constitutive mixotroph grouping. Such indirect photosymbiotic relationships in the model might miss any differential climate sensitivities of symbiont and host. Furthermore, the current parameterisation of calcification, spines, and symbiosis will not respond directly to environmental changes, such as bleaching at high temperatures (Edgar et al., 2013) or reduced weight under high $CO_2$ (Barker and Elderfield, 2002). However, relying on parameterisation is common in EMICs (Claussen et al., 2002), as quantitative experimental studies are lacking now to define the trade-offs and benefits. Furthermore, this lack of understanding of trade-off and their change during development currently makes it impossible to model the life cycle, though further development would be the inclusion of size classes other than 190 μm.

Some potentially important trait interactions and physiological variation are not included in the model. For example, the model assumes that the spine and symbiosis are independent. However, observations suggest that foraminifera symbionts are placed along spines during daytime (LeKieffre et al., 2018), increasing the efficiency of the symbiont's photosynthesis due to a higher surface area relative to non-spinose species by avoiding shading.

## 10 Ecosystem model implementation and complexity

Current coupled Earth system and ecosystem models mostly rely on nutrient–plankton–zooplankton–detritus (NPZD) (Keller et al., 2012; Watanabe et al., 2011) or plankton functional type (PFT) (Moore et al., 2001; Aumont et al., 2015) approaches. The NPZD models focus on biogeochemical fluxes and ignore diversity of phytoplankton and zooplankton. In contrast, PFT models explicitly represent plankton functional types (e.g. diatoms, coccolithophores) and size classes (e.g. picoplankton, nanoplankton, microplankton), improving performance in reconstructing observed patterns like Chl $a$ (Quéré et al., 2005) or peak production in oligotrophic areas (Tréguer et al., 2018). Additional traits beyond size, like symbiosis (Suggett et al., 2017) or body extension (Ohman, 2019), play an important role in determining plankton feeding, metabolism, and export efficiency but are often missing in the current generation of coupled models. Trait-based models, such as Darwin (Follows et al., 2007) and EcoGEnIE (Ward et al., 2018), resolve higher plankton diversity by linking key traits with trade-offs (e.g. the allometric relationships for size), allowing a more continuous representation based on physiology (Follows and Dutkiewicz, 2011). This approach enables the inclusion of non-culturable

species or species with limited laboratory data. Uniquely, this modelling approach also allows us to characterise extinct taxa and past geological records with different physiologies and niches.

There is still a debate on whether higher ecosystem complexity is needed (Anderson, 2005; Quéré et al., 2005) as more parameters introduce more freedom and longer run time. However, recent studies highlight the importance of biodiversity in the marine biological pumps (Tréguer et al., 2018). The presence of functional groups like diazotroph can alter the response of primary productivity to global warming (Bopp et al., 2022). Therefore, compared to the simple food web structure in current models, ecosystem implementation is very likely going to improve the future prediction of biological carbon pump and carbon cycle (Wilson et al., 2022) building on novel additions in models of ecosystem complexities such as more functional types, variable stoichiometry, and nutrient co-limitations (Séférian et al., 2020).

## 11 Summary

In this study, we extended the trait-based planktic foraminiferal model, ForamEcoGEnIE, to include symbiosis and spine traits and thereby resolve all main foraminiferal functional groups. Using Latin hypercube sampling, we generated 1200 parameter samples and compared these with three global observational sources: sediment surface coretop, plankton nets, and sediment traps. We assessed the model performance describing biogeographic distributions, and quantifying carbon biomass and foraminifera-derived carbon export. Our global sensitivity analysis shows that the symbiosis and the palatability reduction due to the spinose test strongly influences model performance. Our best set of model parameters successfully reproduces the modern biogeographical distribution of the four foraminiferal ecogroups and produces a global annual mean biomass (0.001 to 0.010 mmol C m$^{-3}$) and foraminifera-derived organic carbon export (0.002–0.031 mmol C m$^{-2}$ d$^{-1}$) close to observations. The two symbiont-barren groups account for 85 % of standing stocks and foraminifera-derived carbon export. The model accurately reproduces the peak time of seasonal time-series observations of foraminiferal biomass and organic carbon flux but performs poorer in seasonal amplitudes, particularly in upwelling regions. These results provide confidence in the model's ability to explore foraminiferal ecology and diversity in the geological record, for example of the last glacial maximum, as well as helping to solve riddles about their ecology in the past. The trait-based framework of the cGEnIE ecosystem provides the potential to extend the model by presenting more traits such as life history and differential calcification rates across groups.

*Code and data availability.* The source codes and data are archived at https://doi.org/10.5281/zenodo.6808760 (Ridgwell et al., 2022). The experimental configuration and the observational dataset can be found in the following directory: genie-userconfig/MS/yingetal.GMD.2022. CE9

*Supplement.* The supplement related to this article is available online at: https://doi.org/10.5194/gmd-16-1-2023-supplement.

*Author contributions.* RY, FMM, JDW, and DNS designed the study. RY, FMM, and JDW developed the model code. RY performed the experiments, data collection, and visualisation. All authors interpreted the data and wrote and edited the original draft.

*Competing interests.* The contact author has declared that none of the authors has any competing interests.

*Acknowledgements.* The authors wish to thank the two anonymous reviewers for their helpful and constructive comments. We would also like to thank Maria Grigoratou for her insights on the model development and Ruby Barrett for proofreading the manuscript.

*Financial support.* This research has been supported by the Natural Environment Research Council (grant nos. NE/P019439/1 and NE/N011708/1), the China Scholarship Council (grant no. 202006380070), the AXA Research Fund (AXA Research Fund Postdoctoral Fellowship) and the Micropalaeontological Society (Angelina Messina grant). TS52

*Review statement.* This paper was edited by Paul Halloran and reviewed by two anonymous referees.

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

**Remarks from the language copy-editor**

CE1    Please note the change throughout.

CE2    Do you mean "rhizopodium" throughout? Please check.

CE3    Do you mean "carnivorous"?

CE4    Please confirm the change.

CE5    Please confirm the change.

CE6    Please confirm the change.

CE7    Please confirm the change. Please note that, according to our house standards, italic font is not permitted for R packages. Apologies for the original oversight.

CE8    Please confirm the change.

CE9    Please confirm the section.

**Remarks from the typesetter**

TS1    Please give an explanation of why this needs to be changed. We have to ask the handling editor for approval. Thanks.

TS2    Please give an explanation of why this needs to be changed. We have to ask the handling editor for approval. Thanks.

TS3    Please give an explanation of why this needs to be changed. We have to ask the handling editor for approval. Thanks.

TS4    Please give an explanation of why this needs to be changed. We have to ask the handling editor for approval. Thanks.

TS5    Please give an explanation of why this needs to be changed. We have to ask the handling editor for approval. Thanks.

TS6    Please confirm subscript in "$B_{i_C}$". Is it correct?

TS7    Please give an explanation of why this needs to be changed. We have to ask the handling editor for approval. Thanks.

TS8    Please give an explanation of why this needs to be changed. We have to ask the handling editor for approval. Thanks.

TS9    Please give an explanation of why this needs to be changed. We have to ask the handling editor for approval. Thanks.

TS10    Please give an explanation of why this needs to be changed. We have to ask the handling editor for approval. Thanks.

TS11    Please give an explanation of why this needs to be changed. We have to ask the handling editor for approval. Thanks.

TS12    Please give an explanation of why this needs to be changed. We have to ask the handling editor for approval. Thanks.

TS13    Please give an explanation of why this needs to be changed. We have to ask the handling editor for approval. Thanks.

TS14    Please give an explanation of why this needs to be changed. We have to ask the handling editor for approval. Thanks.

TS15    Please give an explanation of why this needs to be changed. We have to ask the handling editor for approval. Thanks.

TS16    Please give an explanation of why this needs to be changed. We have to ask the handling editor for approval. Thanks.

TS17    Please give an explanation of why this needs to be changed. We have to ask the handling editor for approval. Thanks.

TS18    Please give an explanation of why this needs to be changed. We have to ask the handling editor for approval. Thanks.

TS19    Please give an explanation of why this needs to be changed. We have to ask the handling editor for approval. Thanks.

TS20    Please give an explanation of why this needs to be changed. We have to ask the handling editor for approval. Thanks.

TS21    Please give an explanation of why this needs to be changed. We have to ask the handling editor for approval. Thanks.

TS22    Please give an explanation of why this needs to be changed. We have to ask the handling editor for approval. Thanks.

TS7    Please give an explanation of why this needs to be changed. We have to ask the handling editor for approval. Thanks.

TS8    Please give an explanation of why this needs to be changed. We have to ask the handling editor for approval. Thanks.

TS9    Please give an explanation of why this needs to be changed. We have to ask the handling editor for approval. Thanks.

TS10    Please give an explanation of why this needs to be changed. We have to ask the handling editor for approval. Thanks.

TS11    Please give an explanation of why this needs to be changed. We have to ask the handling editor for approval. Thanks.

TS12    Please give an explanation of why this needs to be changed. We have to ask the handling editor for approval. Thanks.

TS13    Please give an explanation of why this needs to be changed. We have to ask the handling editor for approval. Thanks.

TS14    Please give an explanation of why this needs to be changed. We have to ask the handling editor for approval. Thanks.

TS15    Please give an explanation of why this needs to be changed. We have to ask the handling editor for approval. Thanks.

TS16    Please give an explanation of why this needs to be changed. We have to ask the handling editor for approval. Thanks.

TS17    Please give an explanation of why this needs to be changed. We have to ask the handling editor for approval. Thanks.

TS18    Please give an explanation of why this needs to be changed. We have to ask the handling editor for approval. Thanks.

TS19    Please give an explanation of why this needs to be changed. We have to ask the handling editor for approval. Thanks.

TS20    Please give an explanation of why this needs to be changed. We have to ask the handling editor for approval. Thanks.

TS21    Please give an explanation of why this needs to be changed. We have to ask the handling editor for approval. Thanks.

TS22    Please give an explanation of why this needs to be changed. We have to ask the handling editor for approval. Thanks.

TS23    Please give an explanation of why this needs to be changed. We have to ask the handling editor for approval. Thanks.

TS24   Please give an explanation of why this needs to be changed. We have to ask the handling editor for approval. Thanks.
TS25   Please confirm removal of italic formatting.
TS26   Please give an explanation of why this needs to be changed. We have to ask the handling editor for approval. Thanks.
TS27   Please check throughout the text that all vectors are denoted by bold italics and matrices by bold roman.
TS28   Please confirm that the subscript is correct.
TS29   Please give an explanation of why this needs to be changed. We have to ask the handling editor for approval. Thanks.
TS30   Please give an explanation of why this needs to be changed. We have to ask the handling editor for approval. Thanks.
TS31   Please give an explanation of why this needs to be changed. We have to ask the handling editor for approval. Thanks.
TS32   Please give an explanation of why this needs to be changed. We have to ask the handling editor for approval. Thanks.
TS33   Please give an explanation of why this needs to be changed. We have to ask the handling editor for approval. Thanks.
TS34   Please give an explanation of why this needs to be changed. We have to ask the handling editor for approval. Thanks.
TS35   Please give an explanation of why this needs to be changed. We have to ask the handling editor for approval. Thanks.
TS36   Please give an explanation of why this needs to be changed. We have to ask the handling editor for approval. Thanks.
TS37   Please give an explanation of why this needs to be changed. We have to ask the handling editor for approval. Thanks.
TS38   Please give an explanation of why this needs to be changed. We have to ask the handling editor for approval. Thanks.
TS39   Please give an explanation of why this needs to be changed. We have to ask the handling editor for approval. Thanks.
TS40   Please give an explanation of why this needs to be changed. We have to ask the handling editor for approval. Thanks.
TS41   Please give an explanation of why this needs to be changed. We have to ask the handling editor for approval. Thanks.
TS42   Please give an explanation of why this needs to be changed. We have to ask the handling editor for approval. Thanks.
TS43   Please give an explanation of why this needs to be changed. We have to ask the handling editor for approval. Thanks.
TS44   Please give an explanation of why this needs to be changed. We have to ask the handling editor for approval. Thanks.
TS45   Please give an explanation of why this needs to be changed. We have to ask the handling editor for approval. Thanks.
TS46   Please give an explanation of why this needs to be changed. We have to ask the handling editor for approval. Thanks.
TS47   Please give an explanation of why this needs to be changed. We have to ask the handling editor for approval. Thanks.
TS48   Please confirm reference list entry.
TS49   Please confirm citation.
TS50   Please confirm.
TS51   Please confirm caption.
TS52   Please confirm acknowledgements and financial support sections.
TS53   Please note that only one URL is allowed in a reference list entry. Please provide date of last access.
TS54   Please check all numbers. Are they correct?
TS55   Please confirm DOI.
TS56   Please provide edition, editors (if any) and ISBN/DOI. Please note that the Jarman, 2020, reference has been removed.
TS57   Please provide date of last access.
TS58   Please confirm reference list entry.
TS59   Please provide initial.