# Peer review of "ForamEcoGEnIE 2.0: Incorporating symbiosis and spine traits into a trait-based global planktic foraminifera model"

_Geoscientific Model Development, 2022_

## Author Comment (AC2)

We thank both reviewers for their constructive comments on the manuscript. We have made some major changes following the two reviewers' comments.

- Firstly, we changed the POC export unit conversion (Reviewer #1's 4th major comment) and slightly retuned the model to account for this change. This change generates better model-data comparison than the previous version.
- We also reformatted the text, improving the introduction and the model description as Reviewer #2 requested. The introduction now justifies the focus on the critical gap in foraminiferal model development and introduces better the traits of spine and symbiosis to the readers. The model description update now includes a new figure demonstrating the basic model structure.
- Lastly, we added additional discussions about why we need to increase the foraminifera complexity in a model and the possible limitations in the current parameterization.

These additions have increased the manuscript's length while improving its clarify. We hope the editor agrees with us that this was worthwhile.

We have responded to each comment below. Reviewer comments are shown in bold, our responses in blue and our actioned responses in red (with quoted text in Italics).

**Reviewer #1**

**First, the authors must provide a much more detailed description of the cost function (or M-score) they use to tune the model and assessment of the limitations of the various data sets they are tuning it to. For example, it is not clear how they deal with substantial space-time patchiness in tow data and how they control for seasonal biases therein. It appears to me they may be comparing annual means form the model to relatively instantaneous tow observations which presumably occurred at different times in different places. Such observations would be unlikely to represent the annual mean anywhere with any seasonality. If this is true, I would strongly recommend conceiving of a more robust way to control for seasonality in the sparse obs. If it is not true then what they did do needs to be much more clearly explained. Similar concerns are detailed in Major Comment 1.**

We now provide more details on the cost function in the Section 6.3.

*"We used the Mielke measure, or M-score (Watterson, 1996; Watterson et al., 2014) to quantify the model-data fit (Eqn. 21). This metric is essentially a non-dimensional transformed mean square error (Gregoire et al., 2011; Hemer and Trenham, 2016). The score spans from -1 to 1 with values closer to 1 representing better model performance, 0 representing no predicting skills, and negative values representing negative correlation."*

It is correct that we calculated the cost function using model annual mean with observations, however, we estimated observed annual mean averaging all observations present in each model grid point in time (combining different seasons and years). To assess the observation temporal coverage, we compared the model with the observed seasonality (Figures 9, 10). Combining annual mean cost function with seasonal plots thus allow us to validate the model temporally and spatially. We will clarify this in the comparison section (Section 6.2).

**Second, a much richer analysis/discussion is warranted justifying why the inclusion of increasing foram complexity is useful for resolving large BGC cycles (beyond just being able to resolve foram diversity for its own sake). What large scale BGC processes/mechanisms might be getting missed by not resolving this level of complexity? Are there any metrics by which ForamEcoGenie 2 performs better than its predecessor which can be contributed to improve fidelity of foram diversity?**

We agree with the reviewer that the long-term goal of implementing foraminiferal complexity is to better resolve large GBC cycles. However, this goal is beyond the scope of this current paper.

Foraminifera influence biogeochemistry mostly via the inorganic carbon cycle rather organic carbon cycle, as they have very low biomass. Modelling calcification of foraminifers is currently limited by the lack of understanding of the mechanistic drivers behind carbonate production. Instead, as is common in Earth System Models, we estimate calcification in the model using a fixed rain ratio to estimate the calcite export from the organic carbon export of the foraminifera.

Increasing the complexity of foraminiferal traits allows us to 1) capture their different niches, 2) improve the comparison to observations, which resolve this degree of ecological complexity, and 3) capture foraminifera biogeographic and ecogroup change in response to the environment. Having a more complex foram model also provides a powerful tool to

compare with past events and possible impacts of future climates. These additional benefits had relatively minor impacts on the performance of the ecosystem properties in EcoGENIE (Figure 12).

We have clarified our study's focus, resolving the ecology rather than the biogeochemical role of planktic foraminifera (Line 87). We have also added a more general discussion about the importance of model ecosystem complexity in Section 10, second paragraph.

**Third, some additional discussion point warrant consideration. For example, I am curious if the model, and sensitivity study in particular can provide insights into the seemingly stark dichotomy between observed foram biomass (very low) and observed foram export production (very high). It would also nice to add a more focused discussion of the mechanisms (i.e. parameterized physiologic trade-offs) that lead to the emergent foram distributions.**

We thank the reviewer for questioning the cause of discrepancies between distribution patterns, biomass and export. To disentangle the cause, we plotted the histogram below showing the model parameter values that are associated with low foraminifera export production (links to the cluster with high M-scores). In brief, the foraminifera cell leading to high scores has higher probabilities falling in the 100-200 µm range. We interpret this as zooplankton in this size tend to resemble the observed foram's spatial distribution (Figure S5 below). As for the other parameters, the symbiont size is small (0-0.01 times the host size) so that their high nutrient affinity supports foraminiferal survival in oligotroph gyres. The calcification respiration peaks at 0.02 mmol C/d to achieve better comparison with the observed low standing stocks.

Thanks to this analysis, we were able to identify a mistake in the data processing, causing artificially low biomass to export production ratio. We previously converted observations of foraminifera export into the incorrect unit. Fixing the unit conversion of the foraminifera export and retuning the model now provides modelled magnitude for biomass and export more consistent with observations. Despite this improvement, our main concern of underestimating these two metrics because of the limited temporal and spatial coverage in the data still remains. The real biomass is likely higher than in the reported observations (if all seasons were equally sampled) to match the high export production.

We have added a histogram (Figure S3, shown below) to provide an overview of the optimal parameters (indicating the emergent distribution). We modified our unit conversion and present now the retuned model. We also provide a discussion in the 7.1 Section and a more distribution-relevant discussion in 7.2.

[Figure]

Figure S3. Histogram of the optimal parameters (linking to Cluster "A" in Figure 2) associated with low foraminifer annual mean export production (< 1 mmol C m$^{-2}$ d$^{-1}$) and relatively high relative abundance M-score (>= 0.45). Parameter abbreviations are as follow. cal, calcification; mort, mortality; red, reduction strength; palat: palatability; respir, respiration; a, autotroph; h, heterotroph. ss, symbiont-obligate spinose foram, sn, symbiont-facultative non-spinose foram.

[Figure]

Figure S5. Biogeographical distribution of zooplankton biomass in 190 μm

**Finally, the authors need to be much clearer in their nomenclature throughout, both in figures and text it is often unclear what set of observations and sometimes what metrics are being referred to. Moreover, it is often unclear what dimensions/scales variables are being average over.**

We thank the reviewer's careful observation on the caption/terminology. We have made corresponding changes in figures and text as the reviewer asked (see responses to the minor comments).

**Major Comments**

**Model Evaluation and Cost Function.**

**Primarily, it is not clear how the time dimension is incorporated into the evaluation of model skill and/or if model and obs are being compared on consistent time scales.**

**How do you deal with the different time scales of different obs? The cores samples are presumably treated as averaged across a much longer time scale (certainly averaged across any seasonal signature). However, the tows and traps are measureing things on much shorted time scales. Depending on the method you could probably pretty easily get annual averaged in the trap data but the tows would certainly carry a seasonal signature which could bias the comparison with model means. How can/do you control for this? Although as noted below, it is not actually clear the model metric is the annual mean.**

**Are all M-scores computed on annual averages? If yes, then are they climatologies? And then, is there any accounting for the fidelity of the seasonal cycle?**

**For the obs that don't average out the seasonal cycle is the M-score somehow paired in time between model and obs? Or are there enough obs in all cases for a robust annual mean to emerge (this seems unlikely for the tows)?**

1) Time scale
While the core-top data are already averaged over several decades given bioturbation in surface sediments, we plotted a histogram below to show the sampled seasonality in plankton net and sediment trap studies for demonstration. As the reviewer points out, the plankton net studies barely consider seasonality, and this might be not robust for annual average comparison. We deal with this by directly comparing the model with seasonal time series in Figure 9 & 10. However, we will also clarify the limitation in annual average biomass comparison part (section 7.3).

2) M-score
The M-scores are based on annual averages, mimicking a climatology in the sense that we combine multi-year observations. We did not estimate a cost function for the time series comparison, because (1) the sampling data temporal coverage is too low at most locations, and (2) the number of available locations is insufficient (like those in Figure 10), creating large spatial bias towards specific oversampled locations, (3) the coarse model grid resolution isn't that well suited to resolving seasonal cycles in detail so an annual average is a more consistent comparison with the model.

As mentioned above, we now clearly specify the data processing and model-data comparison in the method section 6.2.

[Figure]

A histogram of sampled month in collected sediment trap and plankton net data. The sediment traps tend to carry seasonal signatures while plankton net not.

**What assumptions justify comparing pre-industrial paleo data for one metric (relative abundance) to very recent anthropogenically forced data for the others (absolute concentration and export)?**

To clarify, we calculated a score for each metric, so did not quantitatively compare different observations. The model is forced with pre-industrial boundary conditions to match the core-top data (relative abundance). So there is possible inconsistency in comparing the top-core data (representative of the pre-industrial state) and water-column observations (plankton net and sediment trap, which represent the current climate). But we assume that such inconsistency is negligible at the first-order level considering (1) the scale of foraminifer living stocks is small and (2) the difficulties of tackling different time scales of those plankton net and sediment trap (from 1970s-2010s). This is clarified in the Line 316.

**Why is the trap data in units of count/m3/d rather than count/m2/day. Presumably the trap POC starts as a volume, but shouldn't that be divided by the height of the trap container to get a flux?**

We are grateful that the reviewer has found this inconsistency. The sediment trap data should indeed be in count/m2/d and not count/m3/d as used in the manuscript. We have now fixed the sediment trap data unit conversion and retuned our model. This updated version improves our results with POC export and biomass more consistent with observations.

We converted all modelled flux units into "mg/m2/d", retuned the model and have updated the text throughout to reflect this.

**Line 312: What is the 'time slice comparison' for which you regridded? I couldn't find the term 'time slice comparison' mentioned anywhere else in the manuscript? Was there any re-gridding for the other comparisons?**

We apology for this confusing term. We meant "annual average" and have rephrased it as is.

**Describe a little more specifically how the median absolute deviation measurement ensures 'close to reality data'.**

Using a median absolute deviation measurement improves the model-data comparison when the data are sparse. This uneven distribution results in a few data points with high biomass/export variability having a large effect on the overall scoring. Such high variability can be seasonal or caused by any other local changes to the environment such as storm events which is not resolved in the model.

Because we have now better model-data comparison of export and biomass, we do not need to include a median deviation and have removed its mention in the manuscript.

**Is it necessary to discard species with less than 3% abundance when you are aggregating species into function groups anyway? Considering there are 50 some species I would assume there are quite a few beneath that threshold in each functional group and thus integrate to a non-trivial proportion of the group. It would be good to quantify how many were discarded in each group (in some average sense), or perhaps see how including them influence the M-score of just the optimal parameter set.**

We did not exclude the entire taxa because of rare abundance in some places. We only excluded the occurrence of taxa with less than 3 % of the total assemblage (traditionally 300 specimens would be counted) because the statistical and taxonomical accuracy of these rare occurrence counts is too low. Foraminiferal assemblages have very uneven distribution with a long tails of rare taxa with one or two counts, for which the taxonomy is often less certain than most of the over 30 more dominant taxa. Therefore most of the assemblage is represented by our approach.

**Is the POC flux just separated into that from just Foram groups are all POC?**

Almost exclusively POC flux in the paper reflects the foraminifer-derived bulk POC flux. The only exception is the figure in comparing to prior model versions (Figure 12 in Section 8)

**Cite the figures in which the distribution of each observation is included.**

We assume that the Reviewer refers here to the relative abundance distribution figure.

We have added citations to the subplots in the result section.

**It would be interesting if there was some discussion on the similarities and differences of the parameterization of each Ensemble cluster of Figure 2 (A-E).**

Similarly to Reviewer #1's third major comments, we picked the best cluster (Cluster A in Figure 2) to show the trends in the parameters values with model success in an histogram (Figure S3). We repeated the same analysis (Figure S4 showing below) for the parameters associated with negative M-score (Cluster D in Figure 2). Their distributions closely contrast with the one for the high-score parameters, especially for the first four general foraminifer parameters (i.e., calcification trade-offs and foraminifer size), indicating the important role of foraminifera size and calcification in scoring spatial distribution as they influence all the functional groups.

[Figure]

Figure S4. Same histogram as Figure S3 but associated with negative relative abundance M-score (<=-0.3, proxy of Cluster D in Figure 2).

**It is often ambiguous throughout when biomass and export is being integrated across the whole ecosystem or just forams. Please err on the side of redundant clarification for this as it gets a bit confusing as is. For instance in Figure 3 you look at 'ecosystem biomass' which I assume is integrate across all plankton but also look at POC export which I assume here is integrated across the ecosystem but there is no way to tell from looking at the figure label. Additionally, using consistent use of POC flux and POC export would help (unless you mean different things?) Similarly, it seems like biomass is sometimes referred to as 'living biomass' and sometimes just biomass. Does this mean I am to assume biomass = living biomass + POC?**

We apologise for the confusion. Here "POC export" and "POC flux" refer to the same quantity, and likewise "biomass" and "living biomass" are refer to the same quantity.

We now use these terms consistently throughout the revised manuscript.

**You should define the export depth horizon somewhere else other than the caption of Figure 6. Further there should be some mention of what depth horizon the traps are at. At least on average.**

We agree it is not clear enough in the model-data comparison description.

We provide now more details about the export production depth (80.8m), the average trap deployment depth (-1960 m) in Line 325 and after.

**Section 4.3: Again, it is not clear what time scale you are comparing these on. Are the model distributions global means? And the net tows relatively instantenous points in time? Why would we expect these values to be related as there is presumably some seasonal cycle? Presumably, there is something left unexplained that justifies the comparison, but if not I don't think this a particularly useful metric to assess model skill as it does not appear to be comparing the same thing.**

We addressed these issues in the above responses about net, trap and sediment samples. As mentioned above, the traps/net tows are relatively instantaneous, but we show both annual average and time series comparisons.

 We changed the section headers for sections 7.3 & 7.4 to include "annual average biomass" and "annual average export".

**It would be useful to provide some context on what a good M-score is. In section 4.3 you argue the M-scores are close to zero thus demonstrate the models *inability* to recreate living biomass concentrations; however, every other metric is also closer to 0 than 1. Is that acceptable? Further, does a negative value indicate an inverse correlation or just a worse overall bias? It seems odd that the biomass score is always 0 and never negative unless 0 is some fundamental limit which models with poor skill approach? But then what does a negative value indicate? A strong inverse correlation between model and obs?**

The M-Score for biomass, even with the changes we implemented following the reviewer's comments about unit conversion (see above), still has a relatively low value. This low M Score is despite a good agreement between global annual mean biomass data and model output, both in terms of geographical distribution and global mean range. We tested a method using the geospatial information of the observational points to match nearest model grid (i.e., point to grid) and calculated normalised root mean square. While this method results in a higher score than the grid-grid method we choose, such an approach is also giving undue weight to specific locations where the data is concentrated and therefore creating its own bias. We added text in section 7.3 to explain our approach and its limitations (copied here in Italic text).

We added more description and references to M-score in section 6.3 (see answer above).

*Line 915: "The skill score, however, does not capture this good mode-data fit. This is mostly caused by regridding the data points into model grid resolution. The plankton net data are spatially concentrated in North Atlantic, North-western Pacific, Arabian Sea, and Indian*

*sector of Southern Ocean. Under such circumstance, re-griding causes sparser data and makes skill score sensitive to several outlier grids. Therefore, the insufficient data is likely the primary reason of low scoring in biomass."*

Comparison to Prior Model Iterations:

**The second paragraph of Section 4.1 and Figure 3 touch on how the optimal foram parameter set for EcoGenie 2 compares to previous iterations of the model, but I think this matter warrants considerably more attention.**

**Presumably, the reason for increasing the complexity of a BGC model is to include mechanisms necessary to accurately resolve larger scale carbon and nutrient cycling such that they respond realistically to environmental/climatic perturbations. That is to get things *right* for the right reasons rather than overtunning models without the right mechanisms. So I am curious how this addition improves the performance of the model w/r/t global bgc cycles that might lead us to believe it can offer more accurate predictions that justify its higher cost (computationally and in terms of parsimony). I am thinking about questions like what conditions favour foram groups that transfer carbon to depth or into higher trophic level more efficiently and do we expect climate change to shift that underlying balance in a meaningful way? At a minimum I think some discussion on this front is warranted. But preferably, it would be nice to see some further quantitative comparison of what aspect of global BGC cycling are improved relative to prior, simpler, but computationally cheaper, runs.**

The reviewer raises some very interesting points on the link between increasing model complexity (i.e., functional ecology) and the fidelity of the modelled biogeochemistry. Adding foraminifera will impact two key biogeochemical fluxes: POC and $CaCO_3$ fluxes. We expect a minor impact on POC fluxes because foraminifera only contribute a small fraction of the total plankton biomass. However, associations with the dense $CaCO_3$ test, e.g., ballasting (Wilson et al., 2012) may alter this assumption.

The $CaCO_3$ fluxes of foraminifera tests is likely to impact biogeochemistry as foraminifera are estimated to contribute 23-56% of the total carbon flux (Schiebel, 2002). However, our model does not include an explicit representation of calcification as explained earlier. Secondly, our model does not include other major calcifying groups such as coccolithophores or pteropods (Daniels et al., 2018; Buitenhuis et al., 2019). Therefore, in our model the impact of $CaCO_3$ fluxes on biogeochemistry is limited to the dynamics of productivity via a fixed rain-ratio.

For these reasons, we have chosen not to expand on a quantitative comparison of biogeochemical variables. We have instead re-focussed the manuscript on the plankton ecosystem and associated fluxes such as productivity. We have removed text justifying the development of the model for improving biogeochemistry and have retained text reflecting the impact on biogeochemistry in the discussion (section 8) as a direction for future research.

**At a minimum I would like to see what happens to NPP relative to previous iterations? It is somewhat surprizing that you could achieve similar model skill after adding 3 new tracers without having to tune the parameters of the original model.**

We have now added the NPP comparison in section 8 and figure 12. This analysis shows minor change of POC export with similar geospatial pattern and a small reduction mainly in

the subpolar regions. The similarity of model skill with prior models is likely because the biomass of foraminifera functional types is so small as mentioned before.

**Structurally, with this expanded analysis I think it would flow better if you first describe the skill with which the optimal parameterization of ForamEcoGenie 2 recreates the obs (i.e. Sec. 4.2-4.4 and Figs. 4-6). Then go on to discuss how include accurately resolved foram PFTs changes the overall ecosystem variables in the broader bgc model compared to previous iterations of the model (i.e. Fig 3 and the end of Section 4.1).**

Following the suggestion, we restructured the result section. Now it follows the "parameterisation result -> comparison with observations -> comparison with prior models" route. We also emphasize that we do not focus on how the inclusion of foraminifera diversity changes the nutrient cycles. This is an important direction but outside the remit of this study due to above mentioned limitations.

Additional Discussion

**Discussion of model utility: Per above, can you quantify, or at least more deeply consider, how the added complexity of four foram groups could help BGC models improve large scale nutrient and carbon cy**cling?

We added a section specifically in comparing with prior models (section 8) and a more general discussion (section 10) to discuss how the increasing complexity of foraminifer/ecosystem is necessary.

**Discussion of low biomass and high export: The observations of such low biomass and high export are striking. Especially since the model seems to need much higher biomass to match observed export. A deeper discussion of this could be quite interesting. Could it be a bias in the obs? Nets and traps (especially those that are decoupled in space and time) have plenty of sources of error. Alternatively, what can we learn from the model about how this might be possible from an inverse modelling perspective. Can you identify parameter sets that lead to similar results? What are those parameters? I would assume very low vulnerability to grazing and very high mortality could create such an outcome by preventing recycling and increasing export efficiency. It might also be interesting to look at export efficeincy for forams explicitly. Depending on if there are any interesting findings this may be more suited for a subsection of Res**ults.

See our comments above in relation for many aspects raised here. As for the reason of low biomass and high export, we agree that the reviewer's suggestion of bias in observation is the source of error. In the modified manuscript, we use the correct unit conversion of sediment traps data as suggested by the reviewer and retuned the model as explained above. The biomass and export now compare well with net tow and traps in terms of annual average values. Because the model is calibrated against traps which were deployed in deep waters (average ~2km), and plankton net which likely did not capture the high production season (Line 545).

**Discussion of physiological trade-offs: More discussion of how the assumed (ie parameterized) advantages and disadvantages of each group lead to their emergent distribution would be interesting and warranted.**

We addressed this question above when it was raised before.

*"We have added a histogram (Figure S3, pasted below) showing the optimal parameters (indicating the emergent distribution) and given a general discussion in the 7.1 Section and a more distribution-relevant discussion in 7.2. In brief, the foram size needs to peak in 100-200 um to resemble their prey distribution in high latitudes with abundant nutrients. The symbiont size is small (0-0.01 times the host size) so that their high nutrient affinity can help foraminifer survive in oligotroph gyres. The calcification respiration also peaks at highest bin to achieve better comparison with the observed low standing stocks."*

**Figures and Tables**

**Figure 1.**

**This is redundant with Figure 4, column 2, no? I see how it is useful in an introductory context and definitely needed in Figure 4 for comparison, however, I think you could remove it here and just reference Figure 4 where required. Especially if you are tight on space.**

Yes, they are the same plot. We removed the first redundant one.

**Figure 2.**

**Is the export production shown on the right the globally integrated total foram value used to calculate to the M-score for POC flux? Or is it the total ecosystem POC flux and the former just forams? This is an example of where carefully labelling on what is actually being integrated/averaged is so important.**

**Clarify if each column is the sum of M-scores for all 4 groups with a maximum of 4 (rather than 1) to be transparent that even bright red values are really quite low.**

**Column three should be labelled 'Relative Abundance', not 'Abundance', no?**

**Can you add a column showing the total M-score?**

**Can you highlight the parameter set you chose as optimal?**

The export production shown on the right of the figure is the global annual mean, which is averaged for four foraminifer groups. It shows the common feature of best cluster, i.e. the low export production.

We clarify the summed M-score in the figure caption, abundance is changed to relative abundance; we added the total M-score in the fourth column.

We find it hard to highlight the "best" parameter as this figure, as it summarises 1200 model runs in one plot.

We added some histograms in the supplementary information (Figure S3) showing our best parameter set.

**Figure 3:**

Is there any reason not to show columns 2 and 3 as percent deviation from EcoGENIE such that the bias (relative to EcoGENIE) can be compared across all metrics consistently.

Regardless, it would be useful narrow the colorbar for biomass and POC export as to discern the distribution.

I encourage adding an additional row for NPP.

Clarify in labels and caption that these are ecosystem integrated values, not foram integrated. For example, there is hard to tell if there is a difference between 'POC export ' here and 'POC flux 'in Figure 2, but I believe they are very different variables. I also can't figure out if the you mean something different between 'flux 'and 'export'? If not, pick one and stick with it. Otherwise please clarify throughout.

Potentially move to after Figs 4-6 following my suggestion to shift discussion of model-model ecosystem level comparison to after the model-obs foram level comparison

Thanks for pointing out the importance of this figure.

We have added an NPP row; changed the anomaly to ratio, changed the "POC export" label to "ecosystem POC export"; moved the figure to the latter position and have some discussion about the potential reasons.

**Figure 4:**

"Model relative abundance of each group are calculated based on POC flux rates" – Huh? Is this a typo?

Here and elsewhere, I think the column 1 header should be ForamEcoGENIE 2 to distinguish it from the previous iteration (as in Fig. 3)

Change 'mean 'to 'global mean 'for clarity.

Consider moving the M-score to the row heading on the left, just after the functional group. I think this would be clearer as it is a function of both model and obs and then the heading for each column would be identical (the global mean)

We are sorry for confusion in terminology. Our model estimates relative abundance based on biomass or POC flux, and biomass and POC flux are highly correlated. We have removed this sentence in the caption and added one sentence in Line 378 to state this.

For the figure, the column title of this figure is now ForamECOGENIE 2 as suggested. We also changed the subplots title to global mean, added unit and moved the M-score to the left with functional group name.

**Figure 5:**

I understand why you have overlaid the obs as there are many less data points than in the case of Figure 4. However, I think it would be clearer to present Figes 4-6 in a consistent way, with the model output on the left and obs on the right. Even though there are sparse obs for the other metrics I think this would be easier to compare and better communicate

that the obs are in fact sparse (which is an important point). Further, it would help the reader get their head around all three if they were organized consistently.

Include units and labels for what I assume is the global mean in the header.

Include M-score here too, as in Fig 4. Ideally in the row headers as suggested above

We accepted the reviewer's suggestion to separate the model and data into two columns as the relative abundance figure. The others are same as Figure 4.

**Figure 6:**

**Same comments as Figure 5.**

**Are these units right? Shouldn't export (a flux) be /m2 not /m3 as in Figure 3 and 8?**

The POC export unit is now converted as previously replied.

**Figure 7**

**Are the units of panel b) correct? Shouldn't a flux be /m2 not /m3. Or is there some distinction in the flux, flux rate, and production rate I'm missing?**

**Headings for c) and d) appear wrong. I think c) should be 'globally integrated biomass 'not 'production 'and d) something like 'globally integrated export production 'not POC production rate. I'm positive what 'POC production rate 'means (NPP I suppose?) but I think you are talking about export, no?**

The POC export unit is changed, and the heading is "globally integrated biomass" and "globally integrated EP".

**Figure 9/10**

**Be clear about what obs are being used in each. Presumably tows in 9 and traps in 10, but mention this explicitly in the caption.**

**What do multiple obs data points for the same functional group at the same site during the same month mean?? If these are different species I would integrate them into their corresponding functional groups as done for the M-scores.**

**Minor, but maybe make the model v obs legend in grey rather than blue so that it isn't visually associated with a specific functional group.**

We changed the observations in the figure caption to "plankton net" and "sediment traps."

**Tables**

**Table 3**

**Why is Biomass zeros across the board? I understand it is poorly resolved but being all uniformly 0 and never negative seems odd? See comment above on clarifying interpretation of M-Scores.**

**Caption should read 'M-Score from best model run (or optimal parameter set preferably, per other comment).'**

**Why not include the total M-score (col sum + row sum) as this is ultimately used to decide which parameter set was optimal, no?**

We modified the table. However, we suggest that the low M-score is caused by the low-resolution data (Line 421 and after). The caption is changed to "The distribution of M-scores across foraminiferal groups from the optimal parameter set" akin to other parts of the paper where we now replace "best run" with "optimal parameter set". We have provided the column and row sums.

**Minor Comments**

**Trait Based Model Description.**

**I think it would be useful to have some more introductory discussion on the difference between species-based, PFT-based, and trait-based models, as you often reference species-based models as a foil. .... However, I am not clear if, without the allometric parameterization, there is anything fundamentally different between PFT and trait-based BGC models. Both seems to cluster myriad species into functional (or trait-based) groups and resolve them separately. The difference seems to be just the resolution of the groups (e.g. how many size classes) and how their parameters are related. Further, I think it could be argued that very few BGC models are truly species-specific, but rather, at least implicitly, are averaging over many particularly species. Is there something else essential I am missing? Either way, it would be useful to include a paragraph introducing the differences (similar to the broader intro to BGC model in Ward et al).**

We thank the reviewer for their discussion on the difference of multiple model types. In brief, the trait-based models allow the size spectrum to be continuous, using allometric relationship to determine physiological processes. Using this approach, higher size diversity can be resolved while keeping parameters at a minimum. In addition, PFT models rely on lab data to fit derived growth rates, while trait-based models can be developed and applied to taxa which are challenging to culture to derive physiological understanding like foraminifers. Finally, as pointed out by the other reviewer, PFT and trait-based models overlap, like here defining foraminifera 4 functional groups.

We added a new section (section 10) to introduce the different types of ecosystem models introducing NPZD model (not species-based models), PFT based models, and trait-based models and their strengths.

**Line 1-35: Do coccolithophores and pteropods perform worse as paleoproxies? Mostly, I'm just curious.**

Yes. Isolating individual coccolithophores to monospecific analysis, given their size of a few microns, is challenging. Their organic remains are used as paleo proxies very successfully, though. Pteropods are rare and have a much lower preservation potential as their shell is formed by aragonite a form of calcite which is much less stable, resulting in a more limited use of this group for palaeoproxies.

**Line 60: You have a sentence introducing the 'trait 'of 'symbionts 'and its prevelance. It would be useful to do the same for 'spines 'up top here. Perhaps both following the next sentences. i.e. 'foremost trait is calcification… but spines and symbionts are two more important ones… then sentence on prevalence and definition of symbionts… and sentence on prevalence and definition of spines"**

Thanks for the advice. We added a new sentence (Line 69) to connect the two parts of this paragraph.

**Line 64: Define what 'core-top 'data is.**

We have changed the core top to "foraminifer sediment core-top census data" to make text clearer.

**Line 68:** "spines extruding from the test". Define what the test is?

We changed this sentence to "spines extruding from the calcareous test" and add a note at the second sentences of introduction (Line 25) to explain that test is the synonym of shell.

**Line 111:** Describe the cell quata/carbon quota here a little more explicitly. You focus on how it varies with size but its fundamental role (to vary stoichiometry I think?) is not clear.

We have re-structured the paragraph to highlight the variable stoichiometry and how it influences the nutrient uptake rate (second paragraph in section 3.2)

**Eq 5. Does V stand for Volume and nutrient uptake? If so, change one.**

Thanks for pointing out. We now use $\mu$ to represent uptake rate.

**Line 150 (and elsewhere): It is a bit confusing to use epsilon in the grazing formulation as the common disk parametrization uses the prey capture rate (typically referred to w/ epsilon) instead of the half saturation coefficient (K) to describe a mathematically identical version of the type II response curve. If there isn't a strong reason to use epsilon for the spine effect, I'd suggest changing it to avoid the confusion.**

Thanks for this advice. We now use τ to represent this coefficient.

**Line 299-301: Can you make this either 1 or 3 sentences. As currently written it sound like there is some inherent reason tows and traps a grouped together separate from cores. But as I understand they are three independent data sets each used to evaluate a different aspect of model skill.**

We have separated the sentence of sediment trap and plankton tows.

**Line 342: What is the difference between "POC export scores" and "showing the closest export rate to observations"?**

We wanted to express the higher score accompany lower absolute export values and have rephrased for clarity (Line 361)

**Line 344: Above you say the relative abundance M-score reaches as high 1.2 but here you say the highest is 0.29. I think up top you're referring to the sum all scores for each group, but this could be clearer.**

Yes, it is the summed M-score. We have clarified this in the text.

**Line 345: Does this prioritization mean that the selected parameter set doesn't actually have the highest integrated M-score. Can you quantify this decision by assigning a weighting metric to each variable?**

We do not weight each parameter but choose better performance of relative abundance over the other two.

We have removed this sentence to avoid confusion.

**Throughout: I think 'Optimal Parameterization 'would be more descriptive than 'best run ' which could refer to differences in forcing, initial conditions, etc.**

We have replaced the "best run" with "optimal parameterisation" throughout the article.

**Line 389:" Although the general distribution pattern of foraminifera living biomass agrees with the observations from plankton nets:" --- Does it really? I would qualify this a little more.**

The modified result compares well with plankton net. The difference is same as previously stated.

**Line 395: Export or net primary production? Or primary + secondary production for mixotrophs?**

The cited reference uses production for foraminifer biomass production.

We have removed this terminology.

**Line 409: Here and elsewhere it would help to be really specific if you are talking global POC export of all foram groups, one foram group or all POC. Additionally, it is not clear if you mean something different between POC flux and POC export. Presumably no, in which case use consistent language where possible.**

We apply "POC export" to avoid confusion and make a clear reference of carbon export from ecosystem or foraminifera only.

**Line 414: You use two different references to cite the same range of CaCO3 export. Was that intentional? If so, why?**

The reference should be both Schiebel 2002 Global Biogeochemical Cycle paper.

We have removed the wrong 2001 reference.

**Line 437: Clarify what you mean by species-species discrepancy.**

We removed the term as we rephrased the entire paragraph.

**Lines 404: Agreed. But how does this all influence your M-scores?**

We changed the result as previously stated; consequently, this sentence was also rewritten.

**Line 433: "The model successfully reproduces the first-order seasonal patterns observed by sediment trap data at a basin scale". Does it? Looking at Figure 9 I cant find one panel with a particularly convincing match.**

We rephrased the statement to clarify that for the seasonal time series part, we compared the peak time. While the model does not resemble the amplitudes of the sediment trap, the peak time in the model is largely consistent with data.

**Section 5: This section on limitations focuses entirely on increasingly complex traits that are not resolved but mentions nothing of uncertainty associated with the parameterization of those included or in the observations to which they are tuned. I think some discussion of the latter two limitations is warranted.**

We added an additional paragraph (Starts at line 540) to describe such limitations.

**Line 486: Be specific here: Foram C export or all all C export? Also when you say global mean do you mean globally integrated? Or are you referring to an inter-annual time average?**

We changed the global mean to global annual mean and added foraminifer-derived before the C export.

**Typos and Other**

**Throughout there is a lot of inconsistent/poor grammar that should be improved for clarity.**

**abstract:**

- "increasing functional trait diversity and expanding their ecological niches
- "focusing on functional traits rather than individual species" should
- "observations from global core-tops, sediment traps, and plankton nets"
- "Our model approximates…, accounts"
- "19% of the global pelagic marine calcite budget which is within the lower"

**Intro:**

- "built an ecophysiology based dynamic model" ->" built ecophysiology based dynamic models"

**I've tried to stopped flagging these (although list a few more below) but the grammar warrants a careful review throughout.**

We thank the reviewer for taking the time to highlight these.

All suggestions have been implemented and the paper will be proof read before resubmission.

**Line 44: This sentence is structured as if the model 'reconstructed 'the future scenarios int the second clause. Perhaps revise to "…and simulated potential…"**

Fixed

**Line 70: "traits…lay down the foundation of a trait based model" is a bit of tautology**

We changed "traits" to "observational studies" (as following).

"These observational studies of how functional traits affect biogeography and trophic activities lay the foundation of building a trait-based model."

**Line 95: extra 'and'**

We have removed the 1st "and".

**Line 174: Section title?**

This is a formatting issue as the section title is shown in last page, not resolved.

**Line 404: is a flux rate 'different then a flux?**

We remove all the "flux rate" term.

**Line 445: Should this be a header?**

Fixed.

**References**

Buitenhuis, E. T., Quéré, C. L., Bednaršek, N., and Schiebel, R.: Large Contribution of Pteropods to Shallow CaCO3 Export, Global Biogeochemical Cycles, 33, 458–468, https://doi.org/10/gjpnzt, 2019.

Daniels, C. J., Poulton, A. J., Balch, W. M., Marañón, E., Adey, T., Bowler, B. C., Cermeño, P., Charalampopoulou, A., Crawford, D. W., Drapeau, D., Feng, Y., Fernández, A., Fernández, E., Fragoso, G. M., González, N., Graziano, L. M., Heslop, R., Holligan, P. M., Hopkins, J., Huete-Ortega, M., Hutchins, D. A., Lam, P. J., Lipsen, M. S., López-Sandoval, D. C., Loucaides, S., Marchetti, A., Mayers, K. M. J., Rees, A. P., Sobrino, C., Tynan, E., and Tyrrell, T.: A global compilation of coccolithophore calcification rates, Earth Syst. Sci. Data, 10, 1859–1876, https://doi.org/10.5194/essd-10-1859-2018, 2018.

Gregoire, L. J., Valdes, P. J., Payne, A. J., and Kahana, R.: Optimal tuning of a GCM using modern and glacial constraints, Clim Dyn, 37, 705–719, https://doi.org/10.1007/s00382-010-0934-8, 2011.

Hemer, M. A. and Trenham, C. E: Evaluation of a CMIP5 derived dynamical global wind wave climate model ensemble, Ocean Modelling, 103, 190–203, https://doi.org/10.1016/j.ocemod.2015.10.009, 2016.

Schiebel, R.: Planktic foraminiferal sedimentation and the marine calcite budget, Global Biogeochemical Cycles, 16, 3-1-3–21, https://doi.org/10/bdxfhs, 2002.

Watterson, I. G.: Non-Dimensional Measures of Climate Model Performance, International Journal of Climatology, 16, 379–391, https://doi.org/10.1002/(SICI)1097-0088(199604)16:4<379::AID-JOC18>3.0.CO;2-U, 1996.

Watterson, I. G., Bathols, J., and Heady, C.: What Influences the Skill of Climate Models over the Continents?, Bulletin of the American Meteorological Society, 95, 689–700, https://doi.org/10.1175/BAMS-D-12-00136.1, 2014.

Wilson, J. D., Barker, S., and Ridgwell, A.: Assessment of the spatial variability in particulate organic matter and mineral sinking fluxes in the ocean interior: Implications for the ballast hypothesis, Global Biogeochemical Cycles, 26, https://doi.org/10/gj35bn, 2012.

---

## Author Comment (AC3)

We thank both reviewers for their constructive comments on the manuscript. We have made some major changes following the two reviewers' comments.

- Firstly, we changed the POC export unit conversion (Reviewer #1's 4th major comment) and slightly retuned the model to account for this change. This change generates better model-data comparison than the previous version.
- We also reformatted the text, improving the introduction and the model description as Reviewer #2 requested. The introduction now justifies the focus on the critical gap in foraminiferal model development and introduces better the traits of spine and symbiosis to the readers. The model description update now includes a new figure demonstrating the basic model structure.
- Lastly, we added additional discussions about why we need to increase the foraminifera complexity in a model and the possible limitations in the current parameterization.

These additions have increased the manuscript's length while improving its clarify. We hope the editor agrees with us that this was worthwhile.

We have responded to each comment below. Reviewer comments are shown in bold, our responses in blue and our actioned responses in red (with quoted text in Italics).

**Responses to reviewer #2**

**Major comments:**

**1)   While I agree on the importance of spines and symbiosis to better understand the role of foraminifera in plankton dynamics and key nutrient fluxes in the ocean, I found that the introduction was not strongly motivated to address this gap in knowledge. I think the introduction should be reformulated, stating primary questions and providing more feedback on why spines and symbiosis and important, and importantly, clearly describing the different functional types of foraminifera (instead of focusing on trait-based models; see next comment). In addition, I think the main application of this work goes beyond predicting foraminifer flux. If one wants only to predict nutrient fluxes, statistical models might do a good (and even better) job than mechanistic models. Clearly stating why mechanistic models are important would be helpful. In addition, I think the results should be revisited to provide more mechanistic explanation for the observed patterns: how the tradeoffs implemented here help to explain the model predictions?**

Thank you for highlighting the need for further detail. We were concerned about the length of the paper as well as the details needed and desired in a journal of this focus.

In response to the reviewer, we have reformatted the introduction to address the key modelling gap in the 2nd paragraph and gave an explicit description to symbiont/spine and their function in the 4th and 5th paragraph. The 2nd paragraph addresses the challenge of current foram models in the context of geological record analysis, i.e. not mechanistic and species-based. The 4th and 5th paragraphs now introduce the foraminifer spine/symbiosis and give clear definition of the 4 functional groups.

For the results part (i.e., mechanistic description), we added some more ecological explanations of how we determined the optimal parameter. This part was added in section 7.1.

**2)   The authors put their model forward as a trait-based model. While I think that there is some room for interpretation for what a trait-based model is, I think the authors should be more careful (and specific) here. T…..However, it seems that foraminifera were assumed to have a fixed size in the model and only phytoplankton and zooplankton were assumed to have different size classes. Moreover, neither spines nor symbiosis is implemented using a trait-based approach in the strict sense. Therefore, I don't think the authors should rely too much of their motivation on trait-based modeling as their implementation of foraminifera ecology is mainly based on functional type modeling (and that is fine!).**

Currently most studies define the essence of trait-based models as using measurable functional traits rather than species to link ecosystem function directly. Generally, we agree with the reviewer that our model relies on the parameterisation rather than having several independent physiological processes. However, we argue that there is no fully trait-based ecosystem model in any coupled model. Darwin/EcoGENIE still uses plankton functional types because we lack the ability to map discrete functional types on a continuous spectrum such as body size. ForamEcoGENIE fixes the body size, but it does so as part of the size-based model as a unique "calcifying" zooplankton while zooplankton have a full sizespectrum. While individual size will influence the growth rate and other parameters determined by allometric rules, we lack quantitative observations of developmental changes of growth rate, respiration etc to expand the model and include this approach.

We agree that the phrasing of trait-based model warrants more careful treatment and we have stated this more clearly this now in the model limitation part (Section 9).

**3) The model description requires a through revision as in its current form it is very hard to understand how the plankton ecosystem is being simulated (what are the tracers (i.e. Carbon, nitrogen, phosphorus..), what are the different plankton groups, how they interact, what are the mechanisms modelled and how these are implemented). A schematic could really help the reader here. Even if the model has been published elsewhere, many changes were made here and so a full description of the equations should be given (either in the main text or in the supplement. Please be careful with providing units and descriptions for all model abbreviations that appear in the text. It is also especially hard to follow the functional type modeling for foraminifera and how the new tradeoffs related to spines and symbiosis were implemented.**

We apologise for the unclear model description.

We have provided a new figure (Figure 1) to help demonstrate these four foraminifer types and ecosystem structure/tracers. We also now separate the model description section into 3 parts: cGENIE physics, Ecogenie plankton ecosystem framework, ForamECOGENIE 1 contribution and ForamECOGENIE 2 contribution.

[Figure]

Figure 1. Schematic diagram of ForamEcoGENIE structure, showing the biogeochemical tracers (C, Fe, PO₄) in different colours and plankton populations with various size classes. Physiological processes here include nutrient uptake (red arrows), organic matter production caused by messy feeding and mortality (dashed arrows), and zooplankton grazing (black arrows). A. symbiont-barren

spinose group; B. Symbiont-facultative non-spinose group; C. symbiont-barren non-spinose group. D. symbiont-obligate spinose group. DIC: dissolved inorganic carbon.

**4)  I am not sure I agree with the way authors implement the spine and the symbiosis traits. The implementation of the "spine" trait is done so that spines incur an extra metabolic cost but offers a protection against grazing by decreasing the palatability of foraminifera and also allow them to be more efficient grazers. I am not entirely convinced by this approach for two reasons: i) how do authors calibrate the relative gain from lower palatability and the higher metabolic cost to build spines? A careful analysis to calibrate this tradeoff in a simple model would be very helpful and insightful. Was this provided previously in ForamEcoGENIE?; ii) spines widen the prey availability as explained in the manuscript. Since the authors are using a trait-based model with allometric relationships, why this was not mechanistically represented in the model? Instead, the authors simply decrease the half-saturation constant for grazing, which I don't think translates into the mechanism they are saying they want to model. Meanwhile, symbiosis is represented using the mixotrophy model by Ward and Follows (2016) so that foraminifera can both photosynthesize and consume prey but at a cost of not being as efficient as their specialized competitors. The model by Ward and Follows was developed mainly to describe constitutive mixotrophs, i.e. mixotrophs that possess their own photosystems but can also feed on planktonic prey. Symbiosis is a much more complex process that involves the host to maintain an entire population of their prey inside their cells. Do the authors think their simplified representation of mixotrophy is appropriate to represent symbiosis? What are the limitations of this approach? I suggest to critically consider these points and address throughout.**

1) We calibrated the spine trade-offs using Latin-hypercube sampling of the trade-off parameters to match observation. As for the exploration using a simple model, Grigoratou et al. (2019 Biogeosciences) and Grigoratou et al. (2021 Marine Micropaleontology) tested the trait trade off of calcification and spines. We have included a description of these prior exploratory studies in our last introduction/related method section.

2) It is a good point regarding model implementation and its current implementation of spine and symbiosis. But it is not feasible to generate a fully mechanistic model. Many physiological process descriptions (e.g., Photosynthesis rate-light intensity curve) are built on decades of experimental data, data which is missing for foraminifera. Specifically, we do not fully understand which environmental factors influence the development of spines, nor their full benefit, and their shape can differ between groups with a clear taxonomic but less clear functional difference.

As for symbiosis, foraminifer are not mixotroph as this is a symbiont host relationship. As the trade-offs and benefits of photosynthesis are not well understood, our approach was an approach to develop the skill in the absence of knowledge such as the energy flow between symbiont and host. By regarding foraminifer and symbiont as one entity, we can determine some of the benefits, such as survival in oligotroph areas which was impossible in the previous version. We base this combination on the knowledge that the symbionts will be digested by their host during reproduction (i.e., provide energy) and that they benefit growth (several papers by Allan Be).

We have addressed these model implementation limitations in the section 9.

**5)   Before implementing the four different foraminifera types into a global model, it would have been very helpful to just analyze model behavior in a simpler model, a 0D model that consider idealized environmental conditions. The reader could then better understand how the tradeoffs related to spines and symbiosis play out.**

We focused on implementing the model in a 3-D framework because the basis for some of the model development have already been developed. As stated above, these tests have been performed by in Grigoratou et al. (2019 Biogeosciences) for calcification and Grigoratou et al. (2021 Marine Micropaleontology) for spines. The symbionts have similar implementation as Ward and Follows (2016, PNAS). A key constraint on a model with all four combined traits is the relative abundance across environmental gradients. The EcoGEnIE model provides a consistent physical and biogeochemical framework to predict the spatial patterns in relative abundance along with predicted export production.

**6)   The authors use an extensive dataset to compare their model predictions but I was surprised that the list of foraminifera sp used on this study and the respective functional type classification was not provided in the supplement.**

We added this table (Table S3) in the supplement.

**7)   A strong point of this study is the number of observations that the authors could access to run their ensemble and compare their model predictions against. Although they can represent very well the relative abundance of most of the functional types at a global scale, the absolute biomass is not well predicted by the model. It is also hard to visualize model predictions against observations in Figures 5 and 6 but especially in Fig. 7 (not possible to see the observations in panel a) and in the seasonal plots (Figs. 9 and 10). Also, many of the observations in Figs 9 and 10 do not seem to align with the model. I acknowledge the challenge of comparing model predictions against observations, and that observations are also subjected to error, but I think authors must provide a better way to visualize seasonal patterns and acknowledge model limitations. Perhaps authors can start by comparing model predictions for total nutrients and total foraminifera biomass first since these tend to be easier to simulate than the biomass of different types? #**

We thank the reviewer's point on model-data comparison figures. The inconsistent model-data estimation of biomass/export has been largely solved, as referred to in the very first overview paragraph.

**Response to major comment 3 of the reviewer #1:** *"The low biomass to export production ratio shown in the manuscript it is likely caused by data processing. The modified model version which is calibrated against POC export with correct unit gives consistent magnitude for biomass and export now, though our main concern of underestimating these two metrics because of the limited temporal and spatial coverage in the data still remains. The true biomass is likely higher than observed (if all seasons were equally sampled) to match the high export production."*

As for the visualisation of biomass, export and seasonal patterns, we replotted the annual average map into two separate columns. Figure 7 is related to the outlier measurement method which is now removed in our revised manuscript (related comment copied below). We have also added a new map to the describe the model seasonal peaks in biomass per foraminifera group (Fig 11, shown below), which have a generally good comparison with previous sediment trap study (Jonkers and Kučera, 2015, p.201).

*"To be clearer, using the median absolute deviation measurement can improve the model-data comparison  as the data are sparse, and a few data points with high biomass/export variability will affect the overall scoring. Such high variability can be seasonal or caused by any other local changes to the environment such as storm events which is not resolved in the model.  The new approach suggested by the reviewer of matching observation and model units means this is less of an issue; so we have removed this from the manuscript."*

[Figure]

Figure 11. The peak month of modelled biomass annual time series of each foraminifer group.

**Minor comments**

1)   **Acronyms are used throughout the manuscript and in most of the cases with no previous description, please be sure to provide their definitions.**

Acronyms are now defined in the first place of appearing.

**2)  Model description: start with physics, then describe the biogeochemical tracers and then the plankton components.**

We reformatted the model description part and added the biogeochemical tracers in the 3.1 section now.

**3)  The tradeoffs description requires a reformulation and more detail.**

We have given each trait and trade-off its own section .

**4)  Section 2.4.1 is about respiration, but mortality and palatability are also described in there, confusing.**

We have changed the title to "calcification trade-offs" to be consistent with spine/symbiosis section.

5)  Line 245: Foraminifer predation cost: this description is weak, this is not a good way to frame this, for example, what if everyone is doing this but is doing it wrong?

We do not understand the request of the reviewer here.

We have edited the predation cost part to help clarify any issues.

**6)  Explain up front what the rain ratio means.**

We replaced the "rain ratio" term to P" IC to POC ratio" so that it's clearer for readers.

**7)  Perhaps table 2 could be reformulated to make it clear the differences between the different types, for example, I was surprised that the non-spinose has a Pp that is the same as the spinose type.**

We have reformatted this table and changed some font format, so it should be easier to read and compare.

**8)  Table S1 and S2 should give list of sp and their functional classification.**

As major point 6. we have provided this in Table S3.

**9)  Lines 380: I don't think comparing different models is that informative since they differ in their formulation and goal.**

This sentence is an introduction for readers of the general model performance. But we will not our readers that they are formulated in different ways and goals.

**10)  If foram biomass if overestimated by 8 times, how off do you think the model predicts POC fluxes for each group? Can we still find these estimates robust?**

As major point 7, this has been revisited.

**11) I recommend to provide model code in a repository such as githb and perhaps zenodo.**

We provided model code and relevant data via zenodo as the journal required (readers can find in the **Code and data availability** section).

**References**

Jonkers, L. and Kučera, M.: Global analysis of seasonality in the shell flux of extant planktonic Foraminifera, Biogeosciences, 12, 2207–2226, https://doi.org/10.5194/bg-12-2207-2015, 2015.

Grigoratou, M., Monteiro, F. M., Schmidt, D. N., Wilson, J. D., Ward, B. A., and Ridgwell, A.: A trait-based modelling approach to planktonic foraminifera ecology, Biogeosciences, 16, 1469–1492, https://doi.org/10/gj349g, 2019.

Grigoratou, M., Monteiro, F. M., Ridgwell, A., and Schmidt, D. N.: Investigating the benefits and costs of spines and diet on planktonic foraminifera distribution with a trait-based ecosystem model, Marine Micropaleontology, 102004, https://doi.org/10/gkbn65, 2021.

Ward, B. A. and Follows, M. J.: Marine mixotrophy increases trophic transfer efficiency, mean organism size, and vertical carbon flux, Proc Natl Acad Sci USA, 113, 2958–2963, https://doi.org/10/ggnmm5, 2016.